# The Interplay between Dysregulated Metabolism and Epigenetics in Cancer

**DOI:** 10.3390/biom13060944

**Published:** 2023-06-05

**Authors:** Mahmoud Adel Bassal

**Affiliations:** 1Cancer Science Institute of Singapore, National University of Singapore, Singapore 117599, Singapore; mbassal@bidmc.harvard.edu; 2Harvard Stem Cell Institute, Harvard Medical School, Boston, MA 02115, USA

**Keywords:** cancer metabolism, cancer epigenetics, Warburg effect, cancer hallmarks

## Abstract

Cellular metabolism (or energetics) and epigenetics are tightly coupled cellular processes. It is arguable that of all the described cancer hallmarks, dysregulated cellular energetics and epigenetics are the most tightly coregulated. Cellular metabolic states regulate and drive epigenetic changes while also being capable of influencing, if not driving, epigenetic reprogramming. Conversely, epigenetic changes can drive altered and compensatory metabolic states. Cancer cells meticulously modify and control each of these two linked cellular processes in order to maintain their tumorigenic potential and capacity. This review aims to explore the interplay between these two processes and discuss how each affects the other, driving and enhancing tumorigenic states in certain contexts.

## 1. Introduction

All cancer cells, irrespective of originating tissue, have altered cellular metabolism. Despite being first observed in 1927 by Otto Warburg and colleagues [1], it was not until 2011 that altered cellular metabolism was recognized as a cancer hallmark [2,3]. It is now well established that both liquid and solid tumors upregulate metabolic pathways other than oxidative phosphorylation (OXPHOS) even in the presence of oxygen, a phenomenon termed the “Warburg effect” [4,5]. Cancers have been shown to exhibit increased gluconeogenesis [6], increased glutaminolytic activity [7,8], modified amino acid metabolism [9], increased de novo fatty acid synthesis with reduced fatty acid oxidation (FAO) [10,11,12], and increased pentose phosphate pathway activity [13]. Although Warburg’s original observation identified increased glycolytic rates in cancers, decades of research have shown that cancers near universally show decreased dependence on OXPHOS for ATP production and a greater reliance on other metabolic pathways such as glycolysis, amino acid metabolism, and fatty acid oxidation (to various degrees) to drive ATP production and numerous cellular processes. These observations therefore refine Warburg’s original observations, rather than contradict. Evidence has also accumulated establishing metabolic dysfunction as a strong contributor and predisposing factor to tumorigenesis. It has been shown that prolonged metabolic dysfunction can initiate tumorigenesis via retrograde (RTG) responses, reactive oxygen species (ROS) mediation, and apoptotic or hypoxia inducible factor (HIF)-mediated pathways [14,15,16,17,18,19,20].

The zeal to investigate altered metabolism waned with the advent of gene sequencing in the 1970s and 1980s (the beginning of the “genomics era”) and did not receive significant attention until large-scale next-generation sequencing (NGS) studies began to identify mutations in metabolic genes and pathways [21,22,23,24,25,26,27,28,29,30,31,32]. Decades of research have established causal links between metabolic dysfunction and metabolic disease, neurodegenerative disease, and cancer. These findings have since then reinvigorated research efforts studying metabolic dysfunction in cancers.

While cancers are complex multi-factorial diseases, previous research makes a compelling case implicating metabolic dysfunction as a contributing and/or predisposing factor in many cancers. These observations, however, gloss over the intimate coupling between a cell’s metabolic and epigenetic states. Cellular metabolic states regulate and drive epigenetic changes, while also playing key roles in epigenetic rewiring in response to cell states. Conversely, epigenetic changes can drastically alter cellular metabolics and dependencies. We are therefore only now beginning to piece together the puzzle of how cancer cells meticulously modify and control these two linked and coupled cellular processes in order to maintain their tumorigenic potential and capacity.

In this review, we will be discussing the cellular “powerhouse”, the mitochondria, with a specific focus on the mitochondrial respiratory chain (MRC). The MRC is composed of the five mitochondrial complexes I to V (C-I to C-V). Closely tied to MRC function is the tricarboxylic acid (TCA) cycle. Discussions will then outline how cells respond to metabolic dysfunction and what conditions can arise as a consequence, focusing on MRC and TCA dysfunction and cancer. We will then discuss the epigenome, how it is regulated, and how it is maintained. Discussions will then progress to outline how epigenetic regulators are critically dependent on metabolic cofactors. We will discuss how perturbation of cofactor concentrations adversely affects and alters the function and efficiency of said regulators, with consequent adverse effects on the epigenome. We will then close with a discussion on how closely intertwined metabolism and epigenetic processes truly are, and what targeted therapies are currently in development for cancer.

## 2. The Mitochondrion—Discovery and Structure

First observed in the late 1800s, mitochondria are abundant organelles in all mammalian eukaryotic cell types except erythrocytes [33]. Mitochondrial function was linked to cellular respiration in the early 1900s to 1950s, when the advent of electron microscopy enabled observation of their morphological structure for the first time [34]. Mitochondria are oval or rod-shaped double-membraned organelles with outer and inner mitochondrial membranes (OMM and IMM, respectively) separated by an inter-mitochondrial space (Figure 1) [35,36,37,38]. EM studies have led to the development of two models describing the internal structure of mitochondria, the baffle model and the crista junction model (Figure 1a,b, respectively) [34,39]. The baffle model sees the cristae as random in-folds of the IMM while the crista junction model sees cristae as stacks of independent membranous lamellae. High-resolution EM has shown the internal structure of a mitochondrion to be a hybrid of both models in that cristae are stacks of independent membranous lamellae with random in-folds [39,40].

The mitochondrial OMM separates the organelle from the cytoplasm. Compositionally, it resembles the cellular membrane albeit with a higher lipid concentration, thus facilitating diffusion of lipophilic molecules into the intramembranous space [41,42]. Hydrophilic and small proteins are transported across the OMM through the voltage-dependent anion channels (VDACs), which are ubiquitously expressed in the OMM [43,44,45]. In contrast, the IMM is structurally and compositionally different from the OMM. It is composed of almost 50% more protein and nearly 50% less lipid [41,42,46]. Almost exclusively found in the IMM is the lipid cardiolipin, which has been shown to be essential for normal mitochondrial function [41,42,46,47,48]. The IMM is impermeable to uncharged proteins > 150 daltons in size and contains numerous translocator proteins that facilitate import of charged molecules required for physiological functions (such as adenosine triphosphate (ATP), glutamate, and α-ketoglutarate (α-KG)) [49,50]. Structurally, the IMM forms folds/invaginations named cristae which drastically increase the IMM’s surface area, maximizing ATP production through the MRC [34,36,40].

Under conditions of high energy usage, such as during replication or stress, mitochondria fuse to form extended reticular networks which are disassembled when energy requirements stabilize through a process called mitochondrial fission (Figure 2) [34,51,52,53,54,55,56]. Numerous proteins have been implicated in facilitating fusion and fission events and include myeloid cell leukemia (Mcl1, encoded by *MCL1*), B-cell lymphoma-extra-large (Bcl-xL, encoded by *BCL2L1*), mitochondrial dynamin-like 120 kDa protein (encoded by *OPA1*), mitofusion 1 or 2 (encoded by *MFN1/2*), dynamin-1-like protein (Drp1, encoded by *DNM1L*), phosphatase and tensin homolog (*PTEN*)-induced putative kinase 1 (encoded by *PINK1*), and parkin RBR E3 ubiquitin protein ligase (PARKIN, encoded by *PARK2*) [55,57,58,59,60].

Mitochondria are unique in that they are the only eukaryotic organelle to contain their own genetic material, mitochondrial DNA (mtDNA) [61,62,63]. Mammalian mtDNA encodes two ribosomal and twenty-two transfer RNAs along with thirteen MRC subunit proteins (seven proteins for C-I, one protein for C-III, three proteins for C-IV, and two proteins for C-V [53]). Mitochondria are thus heavily dependent on nuclear-encoded proteins for their biological functions. Transcription of mtDNA has been reported to depend on over 100 nuclear-encoded proteins [53,64], and over 1000 nuclear-encoded proteins must be imported into the mitochondria via translocator of outer membrane (TOM) and translocator of inner membrane (TIM) proteins [50,53,64,65]. Such a high interdependence between mitochondrial and nuclear compartments requires regulation and thus necessitates extensive mitochondrial–nuclear crosstalk [66,67,68]. Research has shown that disruption to this bi-directional communication and/or damage to either compartment triggers compensatory mechanisms in an attempt to restore normal cellular functions [20,67,69,70,71].

## 3. Mitochondrial Cellular Roles

Mitochondria are typically referred to as cellular “powerhouses”, a term first coined in the 1950s [72]. However, mitochondria are also responsible for other cellular functions such as iron–sulfur (Fe-S) cluster formation [73], calcium homeostasis [74], apoptosis regulation [75,76,77], cell signaling, generation of ROS [15,18,50,78,79,80], and generating essential metabolite cofactors required for epigenetic cellular processes (as will be discussed later). Evidence has also implicated mitochondrial involvement in lipid metabolism [81] and autophagosome formation [82].

### 3.1. Oxidative Phosphorylation and Energy Production

Energy production in the mitochondria is performed by the five MRC complexes via OXPHOS. C-I to C-IV form the electron transport chain (ETC), and together with C-V, they make up the five MRC complexes. In a three-step process, energy is released from the ETC and stored as an electrochemical gradient or transmembrane potential across the IMM. This electrochemical gradient drives the conversion of adenosine diphosphate (ADP) to ATP by ATP synthase (C-V) [83]. Generating ATP via OXPHOS has been shown to be up to 15 times more efficient than glycolysis under anaerobic conditions. It has been established that a critically important contributing factor for this increased efficiency is the spatial arrangement of the MRC complexes in the IMM [53].

To date, two models have been proposed for the arrangement of MRC complexes on the IMM, the random collision model and the solid-state model [84,85]. The random collision model proposed that the MRC complexes randomly distribute across the IMM and electron transfer is a random encounter between individual complexes and electron carriers. Contrary to this, the solid-state model proposed that electron transfer is directed from one enzyme to the next within supercomplexes/respirasomes. EM evidence has accumulated supporting the solid-state model and the organization of the MRC complexes in respirasome/supercomplex structures [86,87,88,89,90,91]. These respirasomes are constructed from the ETC complexes (NADH dehydrogenase (C-I), succinate dehydrogenase (C-II), ubiquinol-cytochrome c oxidase (C-III), cytochrome c oxidase (C-IV)) and are arranged in numerous configurations such as complexes I/III_2_ (1*C-I, 2*C-III), III_2_/IV_1–2_ (2*C-III, 1 or 2*C-IV), I/III_2_/IV (1*C-I, 2*C-III, 1*C-IV), and II/III/IV (1*C-II, 1*C-III, 1*C-IV) [89,92,93,94,95,96,97]. Interestingly, a large number of individual C-II units are not found incorporated into respirasomes, possibly owing to the involvement of C-II in the TCA cycle as well as the ETC. The most commonly observed respirasome structure is the I/III_2_/IV configuration, although tissue and species differences have been observed [88,89].

In addition to the ETC supercomplexes, ATP synthase has been shown to form dimer pairs (C-V_2_) which arrange linearly throughout the IMM in a manner separate to respirasomes [89,95].

### 3.2. Iron–Sulfur (Fe-S) Cluster Formation

Iron–sulfur clusters are inorganic cofactors found in many proteins. These cofactors are essential for numerous biological processes such as, but not limited to, enzyme catalysis, electron transfer, and gene expression [98,99,100]. Iron–sulfur cluster assembly has been shown to take place within the mitochondrial matrix, cytosol, and nucleus [98,101].

Within the mitochondria, C-I, C-II, and C-III utilize Fe-S clusters as part of their “catalytic cores” owing to their ability to readily donate and accept electrons. MRC C-I contains eight Fe-S clusters, all of which can be found in the “matrix arm” of C-I. C-II contains three Fe-S clusters while C-III contains only one Fe-S cluster. These cofactors are essential components of these MRC complexes.

### 3.3. Calcium Homeostatic Control

Controlling intracellular calcium concentrations is a critically important task, one in which the mitochondria play a central role [102,103]. Free calcium ions (Ca^2+^) are involved in numerous cellular processes, including being primary and/or secondary messenger molecules or apoptosis-initiating factors [102,104,105,106].

Intracellular Ca^2+^ is drawn into the mitochondria through the VDAC in the OMM and through the Ca^2+^ uniporter in the IMM, as a result of transmembrane potential [107,108,109]. Ca^2+^ efflux is controlled by the mitochondrial sodium/calcium (Na^+^/Ca^2+^) exchanger [110,111]. In doing so, mitochondria control intracellular Ca^2+^ concentrations in parallel to controlling the efficiency of the TCA cycle. This is because three TCA cycle dehydrogenases (pyruvate dehydrogenase, oxoglutarate dehydrogenase, and nicotinamide adenine dinucleotide (NAD^+^)-dependent isocitrate dehydrogenase (NAD^+^-IDH)) are regulated directly and/or indirectly by mitochondrial Ca^2+^ concentration [104,111,112,113,114].

### 3.4. Mitochondria and Cell Death

Four forms of cell death have been documented—apoptosis, necrosis, autophagy, and parthanatos—all four of which require mitochondrial involvement.

## 4. Apoptosis

Apoptosis is triggered once a genetically predetermined set of conditions is reached, hence the term “programmed cell death”. In mammalian cells, two apoptotic pathways exist, the intrinsic and extrinsic pathways. A third apoptotic pathway has also been described involving T-cell-mediated cytotoxicity and perforin–granzyme-dependent killing of cells [115,116,117,118,119] (Figure 3).

The intrinsic pathway has two initiating conditions. The first is the absence of growth factors, hormones, and cytokines, leading to a failure of suppression of pro-apoptotic proteins (bcl-2-like protein 4 (Bax) and Bcl-2 homologous antagonist/killer (Bak)).

The second is by external stimuli such as radiation, toxins, infection, and free radicals. Intrinsic pathway stimulation results in Bax and Bak localizing to the OMM and forming the mitochondrial permeability transition (MPT) pore, releasing cytochrome c (C-IV), second mitochondria-derived activator of caspases (SMAC, also known as DIABLO), and the mitochondrial high-temperature requirement serine protease HTRA2 (also known as Omi) [120,121,122,123,124,125,126]. This results in activation of the caspase-dependent mitochondrial pathway and culminates in cell death [117].

The extrinsic apoptotic pathway, also known as the death receptor pathway, requires a ligand–receptor interaction wherein a ligand, such as tumor necrosis factor-α (TNF-A) or TNF-related apoptosis-inducing ligand (TRAIL, also known as APO2L), binds to a member of the tumor necrosis factor (TNF) receptor superfamily [127]. On ligand binding, cytoplasmic proteins are recruited to bind to the receptor and a death-inducing signaling complex (DISC) is formed (also known as the Fas-associated death domain protein (FADD)) which activates caspase-8 [128,129]. Activated caspase-8 then cleaves BH3-interacting domain death agonist (Bid), which interacts with Bax–Bak, triggering their localization to the OMM and facilitating the formation of an MPT pore [130]. Opening of this pore causes transmembrane depolarization and cytochrome c (C-IV) release and triggers the remainder of the apoptotic cascade [131].

The third form of apoptosis identified is unique to cytotoxic T lymphocytes (CTLs) and natural killer (NK) cells which are capable of killing target cells via the extrinsic apoptotic pathway or via a separate and unique cascade of events. On target cell binding, CTLs release the pore-forming molecule perforin which facilitates transfer of cytoplasmic granules [115,116,132]. Granzyme B, a serine protease which cleaves Bid, is the most important component of these granules [115,116,132,133]. Cleaved Bid then interacts with Bax and Bak, triggering their localization to the OMM where an MPT pore is formed, causing transmembrane depolarization and cytochrome c (C-IV) release and triggering target cell apoptosis [132,133]. Thus, irrespective of which apoptotic cascade is utilized, mitochondria play a central role in the activation of apoptosis.

## 5. Necrosis

Traditionally, four types of necrosis are observed microscopically—coagulative, colliquative, fibrinoid, and caseating necrosis [134]. Necrosis has been viewed as an unregulated form of cell death that occurs due to physical injury of the cell [135]. However, studies have shown that necroptosis, a fifth form of necrosis, is a highly controlled form of cell death that is controlled by “death machinery” (receptor-interacting serine/threonine protein kinases (RIPK)) and is triggered by stimuli such as cell death ligands, infection, DNA damage, and oxidative stress [119,134,135]. Signs of necrosis include cellular content leakage, cytoplasmic granulation, and organelle and/or cellular swelling [136].

An initiating factor of necrosis is ATP depletion [137]. This leads to increased intracellular Ca^2+^ concentrations resulting in mitochondrial calcium overload. Calcium overload then triggers the opening of the mitochondrial permeability transition pore (mPTP), of which the VDAC is a component [65,138,139]. The opening of the mPTP significantly affects the transmembrane gradient, disrupts OXPHOS, leads to mitochondrial morphological changes, and triggers the necrosis cascade [65,139]. Additionally, ROS are also able to trigger necrosis by disrupting lipids, proteins, and DNA, resulting in mitochondrial dysfunction and loss of OMM integrity [139].

## 6. Parthanatos

Parthanatos is a regulated cell death process dependent on the activity of poly ADP-ribose-polymerase-1 (PARP-1) [140,141]. DNA damage due to nicks, breaks, ROS, or ionizing radiation results in overactivation of PARP-1 which consumes ^NAD+^ and depletes ATP stores which potentially inhibits both OXPHOS and glycolysis [140,141,142]. Overactivation of PARP-1 leads to poly ADP-ribose (PAR) synthesis and accumulation, which binds to apoptosis-inducing factor (AIP) on the OMM, facilitating its release from the mitochondria and its translocation to the nucleus [140,142]. Once in the nucleus, PAR induces DNA fragmentation and chromatin condensation which is believed to trigger the caspase-independent cell death cascade [140,141,142].

## 7. Autophagy

Autophagy is the process by which non-essential or damaged cellular constituents are broken down and recycled [143,144,145,146]. Three types of autophagy have been described—macroautophagy (commonly referred to as simply “autophagy”), microautophagy, and chaperone-mediated autophagy [147,148].

Macroautophagy involves vesicle (autophagosome) formation around the target cellular component which then fuses with lysosomes to degrade the contents by acidic hydrolases [148]. Microautophagy occurs when lysosomes directly wrap around cellular contents to be degraded and chaperone-mediated autophagy occurs when chaperone proteins translocate target proteins into lysosomes directly [148]. A specialized role of autophagy, known as mitophagy, has also been described which targets mitochondria requiring degradation and recycling [82,149]. Of note, mitophagy is the only known means by which mitochondria are recycled by the cell [150].

In order for cells to carry out autophagic processes, cells require functional mitochondria. This is because mitochondrially produced ROS are essential inducers of autophagy [147,149,151].

### 7.1. Cell Signaling and ROS

Mitochondria are the main producers of ROS intracellularly as a by-product of OXPHOS generated by C-I, C-II, and C-IV [152,153]. Increased levels of ROS can be generated by the ETC due to subunit damage due to mutation or by ETC inhibition by environmental factors or chemical compounds [154,155,156]. Unsequestered ROS can damage numerous intracellular components such as lipids, proteins, RNA, and DNA [152,157,158,159,160,161]. Increased ROS levels are regularly observed in tumor cells and have been linked to enhancing tumor phenotypes [152,162,163]. It is now accepted also that ROS are intracellular signaling molecules and can modulate cellular functions such as protein function, signaling cascade efficiency, autophagy, or stem cell differentiation, especially in hematopoietic stem cells (HSCs) [147,152,164,165,166].

Modulation of cellular differentiation by ROS is believed to occur via modulation of gene expression of p38 mitogen-associated protein kinases (MAPKs) [167], p53 [168], forkhead box (FOXO) proteins [169], nuclear factor-κB (NF-κB) [170], histone deacetylases (HDACs), and polycomb proteins [171] and through the phosphatidylinositol-4,5-bisphosphate 3-kinase/protein kinase B/mechanistic target of rapamycin (PI3K/AKT/mTOR) signaling pathway [172]. HSCs have been shown to contain low levels of ROS while committed myeloid lineage progenitors contain significantly greater levels [169]. Therefore, controlling ROS production (by controlling OXPHOS) and ROS scavenging mechanisms are critically important for stem and progenitor cells, dysregulation or damage of which could enable intracellular damage or drive unintended premature, or possibly block, differentiation [155,173,174].

### 7.2. Lipid Metabolism

Lipid metabolism occurs intracellularly on the endoplasmic reticulum (ER) [175,176]. Evidence has also shown that mitochondria contribute to this process by synthesizing phosphatidylethanolamine (PE) and cardiolipin [176,177,178]. In yeast, membrane PE is synthesized primarily in the IMM and then transported out to the cytosol. This process involves the protein ubiquitin-specific peptidase 1 (Usp1) [175,176,179]. Cardiolipin, which is found almost exclusively in the IMM, is essential for mitochondrial function and mitochondrial morphology maintenance [176,177,178].

## 8. Mitochondria and Disease

Thus far, we have discussed how mitochondria play significant roles in numerous cellular processes. It is therefore unsurprising that mitochondrial dysfunction due to deficiency, damage, or inhibition has been linked to numerous diseases. We will next discuss the cellular consequences of mitochondrial dysfunction and how mitochondrial dysfunction has been implicated in tumorigenesis and progression.

### Cellular Consequences of Mitochondrial Dysfunction

Mitochondrial function can be affected by numerous conditions such as age [180,181,182], IMM morphological alterations [183,184], lipid concentration perturbations [48], inflammation [185], viruses [186], carcinogens [187,188], hypoxia [189], radiation [188], nuclear and mitochondrial DNA mutation [190,191], and altered expression of the MRC complexes [192,193]. All of these can lead to mitochondrial dysfunction which, phenotypically, can be mimicked by inhibiting MRC complexes using chemical compounds [156].

Mitochondrial dysfunction results in impaired respiration/ATP production which triggers a mitochondrial-to-nuclear signaling cascade of events known as an RTG response (Figure 4) [18,67,70,71,194,195]. An RTG response is triggered when changes to respiration and mitochondrial function are detected, resulting in a metabolic switch from OXPHOS to substrate-level phosphorylation (SLP) by modulating gene expression to maintain the intracellular ATP concentrations required for viability [20,28,69,70,71,194]. So, cells effectively trigger SLP whenever OXPHOS becomes compromised. Although RTG responses have been intensively studied in *Saccharomyces cerevisiae* [195], in mammalian systems, NF-κB/Rel factor responses have been found (through computational homology studies) to most closely resemble an RTG response in yeast [194].

In yeast, activation of an RTG response results in the formation of a heterodimer between the two helix–loop–helix/leucine zipper proteins retrograde regulation proteins 1 and 3 (Rtg1 and Rtg3, respectively) in the cytoplasm which, with the aid of Rtg2, translocate to the nucleus and bind to the GTCAC (R box) sequence in the promoter region of RTG response target genes (Figure 5) [196,197,198]. In mammals, the V-Myc avian myelocytomatosis viral oncogene homolog–MYC-associated factor X (Myc-Max) transcription factors are homologous to the Rtg1–Rtg3 transcription factors [194]. In order to translocate to the nucleus, Rtg3 must be partially dephosphorylated and this is believed to be controlled by Rtg2 [197,199,200].

Negative regulators of RTG signaling include negative regulator of RAS-cAMP pathway (Mks1) and the Bmh1 and Bmh2 proteins [195,200]. Mks1 forms a complex with Bmh1 and Bmh2 (yeast homologs of the 14-3-3 proteins in mammals) and maintain Rtg3 in a phosphorylated state, thus inhibiting translocation [197,201,202,203]. Additionally, Lst8, a subunit of the target of rapamycin (TOR) complex, negatively regulates RTG signaling in a manner distinct to RTG activation by mitochondrial dysfunction [201,202,204].

In mammalian systems, impaired respiration triggers an NF-κB/Rel factor response which causes upregulation of target genes such as Myc and Ras [17,19,20,205]. In a hypoxic setting, however, mammalian systems will also, in conjunction, upregulate the HIFα cascade [206,207,208]. Irrespective of whichever of these signaling pathways is triggered (NF-κB/Rel or HIF1α), the consequent result for the cell is increased Myc expression which enhances ROS production, modulates p53 function, and modulates the expression of genes required for SLP along with other cellular responses [209,210]. (Increased ROS has numerous cellular consequences and will be discussed later [67,211,212]).

P53, a well-established tumor suppressor, modulates numerous metabolic pathways [213,214]. P53 has been shown to modulate the expression of hexokinase and phosphoglycerate mutase, two key enzymes in the metabolism of glucose [215,216]. P53 mediates expression of tumor protein 53 (*TP53*)-induced glycolysis and apoptosis regulator (TIGAR) to inhibit glycolysis [217]. Loss-of-function p53 mutations, therefore, potentially lead to a loss of glycolysis control. Furthermore, overexpression of transmembrane glucose transporters GLUT1 and GLUT4, a common finding in many cancers [218,219], is implicated to be due, in part, to loss-of-function p53 mutation. P53 also activates the PI3K/Akt/MAPK/Ras signaling pathway [220], directly interacts with Bax, Bak, Bcl-2, and Bcl-xl, facilitating apoptotic signaling [221,222,223,224,225], and is regulated by liver kinase B1 (LKB1) [226]. LKB1 knockout mice are hyperglycemic and LKB1^+/−^ mice crossed with p53 null mice show increased tumor incidences and significantly shorter lifespans compared to either mutation/deletion alone [227,228].

Apart from the direct modulation of RTG target gene expression, impaired mitochondrial respiration also triggers numerous compensatory mechanisms through mTORC1 and mTORC2, including, but not limited to, altering expression of genes required for SLP [229,230,231]. Together, these findings describe scenarios wherein mitochondrial dysfunction can lead to increased levels of DNA damage, upregulation of compensatory metabolic and regulatory pathways, and pathology [212,232,233]. Indeed, it is well established that metabolic dysfunction and oxidative stress are linked to metabolic and neurodegenerative diseases and, in certain cases, tumorigenesis and tumor persistence [212,234,235,236,237,238,239].

While metabolic deficiency can arise from direct damage/inhibition of MRC or TCA cycle proteins, it is not limited to these two systems only. As the mitochondria are the central metabolic regulators of the cell, damage, disruption, or inhibition of other metabolic pathways, such as glycolysis, FAO, or transamination, will have cascading effects on the mitochondria and their function by disrupting the availability of metabolites needed for respiration, particularly acetyl coenzyme A (ACoA) (Figure 6). This therefore implies that even dietary intakes, such as high-fat Western diets, can potentially induce chronic changes to cellular metabolic states and function which, over time, can trigger pathological states. 

## 9. Mitochondrial Dysfunction and Cancer Initiation/Progression

Cancer cells, irrespective of originating tissue, have altered cellular metabolics and mitochondria. Over the years, comparative studies between healthy and tumor cell mitochondria have shown molecular, microscopic, metabolic, biochemical, and genetic differences in cancer contexts, while electron microscopy comparisons have shown tumors typically have fewer, structurally altered, and larger mitochondria [241,242,243,244,245,246]. Differential expression of MRC components, which is indicative of mitochondrial dysfunction, has also been linked to numerous cancers [23,247,248,249,250].

With the increasing prevalence of NGS studies, mitochondrial (mt) and nuclear (n) DNA mutations have been identified in numerous cancers including leukemia, breast, lung, liver, kidney, thyroid, ovarian, colon, and brain cancers [21,23,24,25,26,28,31,32]. Unlike nDNA, mtDNA mutations are identified in most cancers with varying prevalence rates (Table 1) [251,252]. Unfortunately, the functional and clinical consequences of the vast majority of these mutations have yet to be elucidated, although many are predicted to impact protein function [252,253]. Of the mtDNA variants that have been studied, consequences are varied with reports of both increased [254] and decreased survival in AML, decreased survival in renal cell carcinoma [255], and enhanced tumorigenicity [256]. Additionally, a survey of mtDNA copy number variation across 22 tumor types identified significant variation in mtDNA copy numbers across the surveyed tumors [257], with additional evidence reporting suppression of MRC gene expression across many cancers as well [23]. Research has therefore been presented highlighting the prevalence of mtDNA mutations, gene expression changes, and copy number variations in cancers, all of which likely lead to metabolic dysfunction and the altered metabolic phenotypes observed in cancers.

As discussed, mitochondrial insufficiency and/or dysfunction, possibly due to mtDNA mutation or copy number alterations, are sufficient triggers to initiate a cellular RTG response. This results in upregulation of genes required for SLP, genes that are commonly reported to be upregulated in both solid and liquid cancers [4,5]. Furthermore, persistent RTG response due to respiratory insufficiency has been shown to cause genomic instability and aberrant growth, two hallmarks of cancers [2,18,70,71,156,205,212,233]. Beyond metabolism, an RTG response also has implications for the cell epigenome, as it radically alters produced metabolite concentrations, which results in modulated or inhibited epigenetic modifier function and efficiency.

Due to the prevalence of the altered metabolic phenotype across nearly all cancers, an important question to raise is whether altered metabolism is causative or a result of transformation. This is discussed below.

## 10. Metabolic Dysfunction and Genomic Instability

In *Saccharomyces cerevisiae*, MRC dysfunction was modeled using small-molecule inhibitors. Sublethal doses of oligomycin (an inhibitor of C-V), antimycin A (an inhibitor of C-III), and potassium cyanide (an inhibitor of C-IV) were applied to yeast so as to ascertain what effects mitochondrial dysfunction has on genomic stability [212]. In doing so, Rasmussen et al. [212] noted an increased prevalence of nDNA mutations in all test groups compared to control cells. Oligomycin-, potassium cyanine-, and antimycin A-treated cells showed a 1.5-, 2-, and 3-fold increase in nDNA mutation frequency, respectively. Furthermore, antimycin A inhibition was found to increase both superoxide (O_2_^−^) and hydrogen peroxide (H_2_O_2_) levels in treated cells. These results showed that MRC inhibition has a nuclear mutator phenotype that is potentially a consequence of increased oxidative stress [212]. Other studies, though, investigated the matter further.

To further characterize the effects mitochondrial dysfunction had on genomic stability, yeast strains lacking their mitochondrial genome (rho^0^ strains) or with fragments of their mitochondrial genome deleted (rho^−^ strains) were created [212]. Both yeast strains (rho^0^ and rho^−^) showed mitochondrial dysfunction and a 2–3-fold increase in nDNA mutation rates over wild-type strains (rho^+^ strains). Decreased levels of ROS were also measured in both strains, potentially due to decreased mitochondrial function. Therefore, the nuclear mutator phenotype observed was independent of oxidative stress-related mechanisms [212]. Rasmussen et al. therefore showed that mitochondrial dysfunction can result in increased nDNA mutation rates and is tumorigenic in nature, involving multiple pathways.

## 11. Metabolic Dysfunction and Aberrant Growth

Apart from gene mutation, mtDNA depletion is a commonly observed feature of many cancers [257,258,259,260]. To investigate the effects of mtDNA depletion on breast cancer tumorigenesis, Kulaweic et al. [233] developed a breast epithelial cell line devoid of mtDNA (ρ^0^ cells). Their results showed that, in vitro, ρ^0^ cells showed a tumorigenic phenotype with enhanced proliferative rates and invasive growth, increased rates of double-stranded DNA breaks, and unique chromosomal rearrangements [233]. Furthermore, ρ^0^ cells in a xenograft SCID mouse model showed a gain of tumorigenicity in normally non-tumorigenic cells and an enhanced tumorigenic phenotype in already transformed cells [233].

To better understand the effects of mtDNA depletion in their ρ^0^ cells, gene expression and pathway analysis was performed. Kulaweic et al. [233] identified 19 regulatory networks with genes showing > 10-fold change in expression in ρ^0^ compared to parental cell lines. One of the highest ranked networks identified had the genomic gatekeeper *TP53* as a focus gene. Kulaweic et al. [233] found *TP53* significantly downregulated in ρ^0^ cell lines and this was recapitulated in primary breast tumors as well [233]. To conclude, Kulaweic et al. [233] suggested that mtDNA depletion plays a role in breast epithelial cell transformation and involves multiple pathways.

Similar observations of increased tumorigenicity have also been documented in mouse C2C12 monocytes and human pulmonary carcinoma A549 cells [17,18,205]. Partial depletion of mtDNA or treatment with metabolic inhibitors resulted in invasiveness of normally non-invasive cells, phenotypes that were reversible on restoration of normal mitochondrial function [17,18,205]. Additionally, mtDNA-depleted cells showed evidence of cellular RTG responses when gene expression was assessed [17,18,205].

The evidence presented therefore suggests that mitochondrial dysfunction can potentially initiate genomic instability, trigger invasive growth, and initiate metabolic and neuronal diseases (Figure 7). However, the question arises, once genomic instability occurs, which phenotype, the mitochondrial dysfunction or the genomic instability, perpetuates the tumorigenic phenotype? To address this question, nuclear-to-cytoplasmic transfer experiments can be interrogated to shed light on this.

## 12. Mitochondrial Dysfunction and the Tumorigenic Phenotype

Healthy parent cells replicate, giving rise to healthy daughter cells (Figure 8a). Likewise, tumor parent cells give rise to tumor daughter cells (Figure 8b). Transfer of tumor-derived nuclei into healthy enucleated cells (cybrids) results in a suppression of tumorigenicity in vitro and in vivo (Figure 8c) [261,262,263,264,265,266,267,268,269,270,271,272]. Cytoplasmic elements can therefore drive the tumorigenic phenotype. To support this, healthy embryonic murine tissue derived from tumor nuclei showed normal phenotypic characteristics despite the persistence of melanoma and brain-associated nDNA mutations, while embryos derived from tumor nuclei did not develop tumors despite persistence of tumor-associated aneuploidy and nDNA mutations [269,273]. Genetic mutation or genomic rearrangements are therefore not necessarily sufficient to induce tumorigenesis alone [274]. Investigations transferring normal mitochondria into tumor cell cytoplasm showed a suppression of tumorigenicity in vitro, suggesting the cytoplasmic elements controlling the tumorigenic phenotypes are indeed the mitochondria [275,276]. Finally, introduction of mtDNA mutations in non-tumorigenic cybrids reverses the anti-tumorigenic effects, giving tumorigenic growth [277]. In 2014, Wahlestedt et al. determined that induced pluripotent stem cells with heavy mtDNA mutation burdens displayed extensive differentiation defects, thus highlighting that mtDNA mutation can lead to aberrant differentiation phenotypes of stem cells [278]. In 2011, Sharma and colleagues observed tumorigenic growth in the presence of C-I mtDNA mutation and a reversal of the tumorigenic phenotype on C-I function rescue. The presented findings showed that healthy mitochondria are able to suppress tumorigenesis despite the continued persistence of tumor-associated DNA mutations and rearrangements. In contrast, tumor-derived mitochondria induced an enhanced tumorigenic phenotype on transfer into healthy cell cytoplasm, suggesting mitochondrial function could be the stronger perpetuating force in continuing a tumorigenic phenotype (Figure 8d) [263,277]. Evidence has therefore been presented from both in vitro and in vivo data implicating mitochondria and mitochondrial dysfunction as key drivers of tumorigenesis and persistence. While DNA mutations undoubtedly can result in dysregulated and non-functional proteins with aberrant function, published evidence shows that mitochondrial dysfunction can have a strong influence on the presentation and continued persistence of tumorigenic phenotypes as shown in numerous in vitro and in vivo models. Future investigations to thoroughly characterize the mitochondrial and metabolic state of cancers would therefore likely lead to significant findings for better and longer-lasting patient treatment as they would be targeting a significant driving force of the tumorigenic phenotype. It is important to clarify, however, that once somatic mutations accumulate within the genome, the combined deleterious effects of both metabolic dysfunction and abnormal/aberrant protein function would unquestionably be significantly deleterious to the state of the cell.

## 13. Mitochondrial Horizontal Transfer Experiments

Over the years, evidence has been presented describing horizontal transfer of mtDNA and/or mitochondria from non-cancerous to cancerous cells which results in restoration of cellular respiration and, in certain scenarios, enhances the tumorigenic potential of recipient cells [279,280,281,282,283]. While such findings initially seem to refute evidence presented that healthy mitochondria can suppress a tumorigenic phenotype, careful examination of these findings only further supports previous evidence and, arguably, partially validates Warburg’s original observations and hypothesis.

Healthy phenotypes are colored green while cancerous phenotypes are colored red.

a—Healthy cells give rise to healthy cells.

b—Tumor cells give rise to tumor cells.

c—Transfer of tumor nucleus into a healthy cytoplasm leads to healthy cells *despite* persistence of tumor-associated genomic mutations and instability.

d—Transfer of a healthy nucleus into a tumor cytoplasm gives rise to tumor cells or death.

For the following discussions, one should take note that the state of a mitochondrion is dependent on four characteristics, (i) mtDNA integrity (i.e., lack of a significant number of mutations and/or insertions/deletions), (ii) sufficient mtDNA copy numbers (which are tissue specific), (iii) the morphological shape of the mitochondrion (both internally and externally), and (iv) the function of internal metabolic pathways such as the MRC and TCA. Cancers show abnormalities in multiple aspects of the mitochondrial state. MtDNA mutations have been linked to, and are causative of, many diseases, including some cancers. MtDNA copy number alterations (both increased and decreased copy numbers) have been associated with numerous diseases, including Parkinson’s disease [284], heart disease [285], and autism [286], and copy number alterations are observed in most cancers [257]. Finally, cancers universally show abnormal mitochondrial morphology and distinct metabolic phenotypes [25,212,287,288,289,290,291,292,293].

In 2006, Spees and colleagues described mitochondrial transfer from human MSCs to A549 lung adenocarcinoma cells. In their studies, Spees et al. reported that the A549 cells with donor mitochondria and mtDNA showed restoration of mitochondrial respiration [279]. While these claims are valid, on examination of the published results, the restoration of the cancer cells’ respiratory capacity was not complete and the recipient clones showed gene expression and gross mitochondrial number differences compared to donor MSCs. Consequently, the mitochondria did not recover to the state of their healthy donor counterparts and could not be classified as “healthy” in all aspects of their mitochondrial state despite *partial* restoration of respiration. Tan et al. [281] made similar observations that mtDNA was found to transfer to donor cell lines, restoring cellular respiration. Tan et al. also observed an enhancement of tumorigenic potential of their cell lines on acquisition of mtDNA [281]. Similar to the findings of Spees et al., examination of the published findings identifies that recipient clones differ in numerous mitochondrial characteristics from their donor cells, ranging from gross mitochondrial number deficits, morphological abnormalities, mtDNA copy number alterations, and gene expression differences [281]. The enhanced tumorigenic phenotype observed can therefore be expected and parallels previously published results that abnormal mitochondria can perpetuate a tumorigenic phenotype [263,277]. These experiments therefore do not contradict the evidence presented previously but, rather, support it.

An additional caveat in interpreting the aforementioned studies relates to the circumstances and direction in which such mitochondrial transfers occurred [280,283,294]. Cho et al. [294] were able to deduce that gross mitochondrial transfer (not mtDNA transfer) only occurred between human MSCs and 143B human osteosarcoma cells that were depleted of mtDNA and did not occur between MSCs and 143B human osteosarcoma clones with mutated mtDNA. These results suggest that mtDNA damage (such as is commonly seen in cancers) may not be a sufficient trigger to instigate horizontal mitochondrial transfer and only extreme levels of mitochondrial damage and depletion will trigger such cell–cell transfers. These results also suggest that neighboring cells can “assist” a cell suffering from extreme mitochondrial dysfunction in order to maintain its respiratory capacity and avoid cell death. Wang and Gerdes [283] observed uni-directional transfer of mitochondria from untreated PC12 rat pheochromocytomas to UV-treated PC12 rat pheochromocytoma cells, which prevented the UV-treated cells from undergoing apoptosis. As discussed, ATP insufficiency is sufficient to trigger cellular death and restoration of mitochondrial respiration by mitochondrial transfer would prevent activation of the apoptotic cascade. Finally, Lou et al. [295] reported mitochondrial transfer between mesothelioma cell lines or between primary human mesothelioma cells but not between cancerous and normal cells [295]. These results therefore suggest that mitochondrial horizontal transfer is context dependent and may not necessarily be a universal phenomenon.

In conclusion, results obtained by mitochondrial transfer experiments, while insightful, do not contradict the nuclear transfer experiments previously described as, despite the transfer of mtDNA and/or mitochondria, numerous aspects of the mitochondria’s state remain altered and the resultant hybrids are not identical to “healthy” donor mitochondria. Furthermore, the assistance of “healthy” cells to donate mitochondria to their metabolically compromised neighboring cancer cells is indicative of cancer cells having defective mitochondria and/or mitochondrial respiration. As this process is predominantly uni-directional (i.e., “healthy” to “cancer”), this potentially supports Warburg’s original hypothesis that cancer cells have compromised mitochondria, a phenomenon that neighboring cells can become aware of (potentially through paracrine signaling) and they can assist in restoring partial mitochondrial function of compromised cells.

## 14. Investigations of Altered Metabolism in Cancers

Many researchers have investigated altered metabolism in cancers [296,297]. Cancers universally show altered metabolic phenotypes and abnormal mitochondria [25,212,288,289,290,291,292,293]. Genomic studies have reported catalogs of mtDNA mutations in numerous cancers [1,21,25,31,252,254,256,257,278,287,296,298,299,300,301,302,303,304,305,306,307,308,309,310]. Despite retaining partial functionality, mitochondria in tumors consistently show mtDNA mutation, structural abnormalities, and diminished functional capacities and tumor cells universally have distinct, reprogrammed metabolic phenotypes which are not observed in non-transformed tissues and cells. To date, no study has yet described a tumor with mitochondria that are functionally, structurally, and genetically indistinguishable from non-transformed cells, to the best of our knowledge. While upregulation of metabolic pathways other than OXPHOS *can* compensate for any potential defects in OXPHOS function, evidence has yet to be presented establishing MRC dysfunction as resultant to stresses within tumor cells and environments.

To establish that MRC dysfunction is consequent to tumor cell genomic and environmental stresses, doxycycline-inducible models have been investigated. While mutation-inducible models have been described [311,312,313,314], the investigations by Ying and colleagues [311] assessed the metabolic state of their model post doxycycline induction. In their study, Ying et al. [311] described a doxycycline–*KRAS*-inducible model wherein activation of a *KRAS* G12D mutant resulted in development of pancreatic ductal adenocarcinoma in nude mice. On activation of the oncogenic *KRAS* mutant, altered metabolic pathway dependencies were observed, including an increased dependence on glycolysis [311]. However, a caveat in interpreting these studies is that doxycycline has known effects on cellular metabolism and considerably shifts cellular metabolism towards a more glycolytic phenotype at commonly used concentrations of 100 ng/mL–5 μg/mL [315]. Therefore, without appropriate experimental controls, use of doxycycline confounds the effects of the mutation under investigation and the true consequent metabolic effects become ambiguous at best. Furthermore, the work of Chang and colleagues [316] showed that doxycycline directly activates the PI3K-Akt signaling pathway, which has been shown to enhance survival and self-renewal in vitro [156,172,316]. Investigations performed using doxycycline-induced models must therefore be carefully analyzed in light of these findings to accurately ascertain the oncogenic effect of the induced mutation and any consequent metabolic changes that are potentially resultant.

## 15. MRC Dysfunction and Cancer

### 15.1. C-I Dysfunction and Cancer

NADH dehydrogenase (C-I) (Figure 9) is the largest complex of the ETC and is a main site of mitochondrial ROS production [317,318]. Of the five MRC complexes, C-I dysfunction is the most common mitochondrial defect observed, leading to disease and cell death [319,320]. Somatic mutations in mitochondrial and nuclear C-I genes have been found in numerous cancers, including leukemia, breast, thyroid, bladder, prostate, colon, pancreatic, and head and neck cancers, and renal carcinomas (select variants listed in Table 2) [31,300,302,303,305,321,322,323,324,325,326,327,328]. Reported C-I variants are predominantly mitochondrially encoded, with fewer data available on somatic nDNA C-I variants. Germline data are more difficult to acquire still, with few publications highlighting their functional significance [329]. Mutations in C-I have also been linked to increased ROS production and increased metastatic potential in mouse tumor cell lines, colorectal cancer cell lines, and HeLa cells [156,304,330]. The degree to which C-I dysfunction plays a role in tumorigenesis and progression therefore varies and depends on both the degree of complex function disruption and tissue type [331,332,333].

### 15.2. C-II Dysfunction and Cancer

Succinate dehydrogenase (C-II) (Figure 9) is part of both the ETC and the TCA cycle. Of the MRC complexes, C-II has the strongest and most well-established link to cancers with succinate dehydrogenase proteins classified as bona fide tumor suppressor genes [236,334,335]. Germline heterozygous C-II mutations have been shown to predispose individuals to cancer, while homozygous mutations in the same genes lead to neuronal diseases [336]. Mutations affecting C-II function and assembly have been linked to hereditary paragangliomas and pheochromocytomas [191], renal carcinoma [337], gastrointestinal stromal tumor [338], and breast cancer [339]. The tumorigenic effects of C-II mutations appear to be due to the accumulation of succinate, which, in excess, inhibits prolyl hydroxylases (PHDs), HIFs, DNA and histone demethylases, and JmjC domain-containing histone demethylation (JHDM) proteins [340,341] (Figure 10), giving rise to a hypermethylation phenotype which resembles that observed in tumors with (for example) somatic IDH mutations [235,236,335,342,343,344].

### 15.3. C-III Dysfunction and Cancer

In addition to C-I and C-IV, ubiquinol-cytochrome c oxidase (C-III) (Figure 9) is a major producer of mitochondrial ROS. Mutations affecting C-III function have been identified in numerous cancers including gastric [345], colorectal [346], ovarian [347], thyroid [348], breast [349] bladder, lung, and head and neck tumors [350]. In bladder cancer, expression of a truncated form of mitochondrial cytochrome b (*MT-CYTB*), the sole mitochondrial encoded subunit of C-III, has been shown to increase cellular growth rates and promote invasion in vitro and in vivo in MB49 bladder cancer cells [351]. This growth phenotype also correlated with increased ROS production and activation of NF-κB2, two characteristics of an RTG response [351]. Additionally, in SV-40-transformed human uroepithelial HUC-1 cells, overexpression of a truncated *MT-CYTB* protein, a truncation previously reported in bladder cancer, resulted in aberrant and sustained cell growth [352]. C-III dysfunction due to mutation can therefore contribute to sustained aberrant growth of tumor cells.

### 15.4. C-IV Dysfunction and Cancer

Cytochrome c oxidase (C-IV) (Figure 9) is the final complex in the ETC. C-IV is unique amongst the MRC complexes in that it is the key regulator and the rate-limiting step of OXPHOS [353]. It also has strong links with apoptosis control [354]. C-IV is also the only complex of the MRC that has known tissue-specific isoforms of nDNA-encoded subunits and has been shown to be modulated by p53 at the mRNA level [353,355]. It has been discovered that p53 regulates expression of cytochrome c oxidase 2 (*SCO2*) [213]. *SCO1* and *SCO2* are involved in transporting copper to the catalytic core of C-IV during assembly and *TP53^−/−^* mice and p53-deficient colon cancer cell lines show decreased oxygen consumption (a key indicator of C-IV function) and ATP generation capacities [213,356].

Of the five MRC complexes, C-IV shows the weakest link between dysfunction and cancer, possibly suggesting a limited tolerance to genetic variation of this complex. Function affecting mutations in C-IV have only been identified in prostate and ovarian cancers [277,357]. In contrast, however, nDNA-encoded C-IV subunits have been found upregulated in leukemia [358] and A549 lung adenocarcinoma cells [359]. In leukemia, increased expression of *COX5A* and *COX5B* correlated with increased ROS [358]. The contribution of ROS to tumorigenesis will be discussed in a coming section.

### 15.5. C-V Dysfunction and Cancer

C-V (Figure 9) is the final complex in the MRC and is the site of ATP generation in the mitochondria. Investigations have observed that C-V forms part of the mitochondrial permeability transition pore (mPTP) [360] and can also be found on the surface of vacuoles [361]. It has therefore been hypothesized that the link between C-V and tumorigenesis could potentially relate to its function as part of the mPTP, with potential links to defective autophagic processes.

C-V subunits have been reported mutated in pancreatic [362], thyroid [348], and prostate cancers [277]. Cybrids containing mutated mitochondrial ATP synthase 6 (*MT-ATP6*) were found to show faster growth rates in vitro and in vivo in xenografted nude mice; a trait which was reversed on reintroduction of wild-type *MT-ATP6* [363]. *MT-ATP6* mutations were also associated with increased levels of ROS, a potential consequence of an RTG response due to mitochondrial dysfunction [360].

## 16. TCA Cycle Dysfunction and Cancer

### 16.1. Isocitrate Dehydrogenase Dysfunction and Cancer

Five isocitrate dehydrogenase (IDH) enzymes are found in human cells. Three NAD^+^-dependent IDH enzymes (*IDH3A*, *IDH3B*, *IDH3G*) are localized in the mitochondria along with the NADP^+^-dependent *IDH2* (Figure 9). *IDH1* is also NADP^+^ dependent but is localized in the cytoplasm. The IDH enzymes catalyze the reversible conversion of isocitrate to α-KG. Both *IDH1* and *IDH2* are somatically mutated in AML predominantly showing hotspot mutations [31,364,365], colon cancer [366], glioma [30], enchondromas [367,368], spindle cell hemangiomas [367], osteosarcoma [369], glioblastoma [27], chondrosarcomas [368], intrahepatic cholangiocarcinoma [370], prostate cancer, and B-acute lymphoblastic leukemia [371].

Somatic mutations in *IDH1* and *IDH2* result in neomorphic enzymatic activity whereby the oncometabolite R-2-hydroxyglutarate (2-HG) is produced [372]. TF-1 human erythroleukemia cell lines incubated with 2-HG became cytokine independent and showed differentiation blockage [373]. 2-HG has also been shown to inhibit α-KG-dependent enzymes such as the PHDs [374], the ten–eleven translocation (TET) family demethylases [375], and the JHDMs [376], which results in DNA and/or histone hypermethylation phenotypes in leukemia and breast cancers [377,378,379]. While 2-HG accumulation in leukemia patients is a direct result of neomorphic IDH mutation, in breast cancer, the accumulation of 2-HG is a result of metabolic dysfunction driven by Myc overexpression [378]. Accumulation of the oncometabolite 2-HG can therefore accumulate in cells independent of IDH mutation due to metabolic dysfunction or hypoxia which has consequent effects on the epigenome [380].

Apart from epigenetic effects, somatic IDH1 mutations have significantly hindered enzymatic function and catalytic abilities [372,381,382]. In glioma, mutant *IDH1* is deficient in performing its oxidative reaction by >80% [383] with an approximate 38% reduction in NADPH generation [384], thus highlighting the metabolic consequences of mutation on its enzymatic and metabolic function.

### 16.2. Fumarate Hydratase and Cancer

Fumarate hydratase (FH) (also known as fumarase) (Figure 9) catalyzes the conversion of fumarate to malate in the TCA cycle. FH has been shown to be an essential regulator of HSC functions, deletion of which results in defective hematopoiesis [385]. Germline mutations in FH have been identified in leiomyomatosis and renal cell cancers [386] as well as paragangliomas and pheochromocytomas [335,387]. Aberrant expression [388,389] and deletion [390] of FH have also been reported in various cancers. Similar to succinate and 2-HG, high concentrations of fumarate have been shown to inhibit α-KG-dependent enzymes such as PHDs and DNA/histone demethylases [235,391,392], leading to hypermethylated DNA phenotypes and aberrant expression profiles.

### 16.3. Citrate Synthase and Cancer

Citrate synthase (CS) (Figure 9) catalyzes the irreversible conversion of ACoA and oxaloacetate into citrate, which can then be exported to the cytoplasm for FAO or processed further by the TCA cycle in the mitochondria. Aberrant expression of CS has been documented in renal oncocytomas, pancreatic ductal carcinoma, and numerous cervical cancer cell lines [192,393,394]. While it is unclear how CS promotes tumorigenesis, the link between CS and cancer could potentially be indirect, wherein CS dysfunction results in mitochondrial dysfunction, resulting in a compensatory and persistent RTG response, giving rise to aberrant growth. Another possibility could be the depletion of citrate metabolite stores, which would result in insufficient ACoA concentrations required to maintain metabolic and epigenetic homeostasis while simultaneously removing a noted glycolysis inhibitor from the cell [395]. However, further investigations are still required.

### 16.4. Aconitate Hydratase and Cancer

Aconitate hydratase (AH) (Figure 9) is an Fe-S cluster enzyme that catalyzes the reversible conversion of citrate to isocitrate. In FH-deficient cell lines, the accumulation of fumarate inactivates the Fe-S of AH and abolishes its enzymatic function, likely triggering an RTG response due to metabolic deficiency [396]. Decreased AH expression has been previously reported in gastric cancers and is also a prognostic marker of disease progression [397].

### 16.5. Malic Enzyme and Cancer

Malic enzyme (ME) (also known as malic dehydrogenase) (Figure 9) catalyzes the conversion of malate into pyruvate and CO_2_. It has been observed that mitochondria isolated from L-1210 mouse leukemia cells showed increased conversion of malate to pyruvate [398]. More recently it was established that knockdown of ME2 diminished proliferation and increased apoptotic rates of K562 erythroleukemia cells in vitro and completely inhibited growth of K562 xenografts in nude mice in vivo [399]. It has also been determined that p53 represses expression of ME1 and ME2, decreasing lipogenesis and glutamine metabolism [305]. These findings therefore link p53 and active metabolic pathways within leukemic cells through modulation of MEs. As deactivating *TP53* mutations can be found in ~10% of AML cases [31], these mutations would enable *TP53* mutant clones to decrease their dependence on OXPHOS, upregulation of which is essential for HSC differentiation, allowing cells to remain in an undifferentiated state, enabling disease establishment and progression, while also making them reliant on pyruvate stores [230].

## 17. ROS and Cancer

ROS produced by the MRC has generally been considered to be deleterious to cells. Recent findings show that ROS are actually important signaling molecules and can modulate numerous cellular functions [147,152,165,166]. Under normal circumstances, ROS production is kept within tight boundaries [400] but tumor cells are often reported as having increased levels of ROS as compared to healthy counterparts [162].

Elevated levels ROS have been shown to affect cell cycle progression and growth factor signaling [401] and promote differentiation in myeloid cells [402] and have been linked to tumorigenesis [403,404]. Excess production of ROS within a cell leads to a state of oxidative stress [405] which is commonly seen in hematopoietic malignancies, including ALL [406], AML [407], and CML [407,408]. While the relationship between ROS, tumorigenesis, and tumor persistence is complex, increased ROS in tumors may support cell survival [409,410], migration [411], metastasis [304], proliferation [412], and genomic instability [413] or even facilitate drug resistance [163] depending on the cancer type by modulating metabolic pathways [408].

## 18. Cellular Consequences of Metabolic Reprogramming

Independent of discussions on the origins of cancer, the universal prevalence of metabolic reprogramming in tumors is indisputable [2,7,414,415,416,417,418,419]. As mitochondria play critical roles in macromolecular biosynthesis, cellular metabolism, and epigenomic control, metabolic reprogramming has consequent effects on these and other cellular pathways affecting cell state. Literature evidence has also been presented implicating the tumor microenvironment (TME) and local inflammation as key factors in supporting and maintaining metabolic phenotypes [420,421,422,423,424], although these too (TME and inflammation) can be controlled by internal cellular metabolic states modulating their surrounding environments [425,426,427].

It has been elucidated that, once glucose is imported into a cell from the TME, cancers utilize large amounts of glucose in a step-wise manner, converting it to pyruvate even in the presence of oxygen [1,188,418,428,429]. Evidence has emerged that a portion of glucose-derived pyruvate is transported into the mitochondria where it is processed for use by the TCA cycle and utilized in subsequent anabolic reactions [428,430,431,432]. Additionally, in tumors which show evidence of defective mitochondrial respiration and/or TCA function, glutamine dependence is observed [430,433,434,435]. Glutamine is becoming recognized as a critical substrate required by many cancers as it is a carbon source for macromolecular synthesis and can also be utilized in producing ATP in a respiration-independent manner [7,430,431,432,433,436,437,438]. Glutamine metabolism is also utilized by tumors for both amino acid and de novo lipid synthesis where required [428,437]. Ultimately, though, tumor cells have metabolic preferences, whether for glucose, glutamine, amino acids, or lipids, that support their growth and proliferative requirements but unfortunately can become potent dysregulators of the epigenome [439,440,441].

In order for a cell to regulate its genome, numerous post-translational modification (PTM) systems are utilized including, but not limited to, phosphorylation, methylation, and acetylation. However, these same epigenetic mechanisms require metabolites such as citrate, pyruvate, NAD^+^, ACoA, ATP, NADH, and flavin adenine dinucleotide (FAD), which originate from metabolic pathways, some of which can be inhibitory for epigenomic regulator function at specific concentrations (such as succinate or fumarate [439]). Metabolic reprogramming resulting in altered metabolite concentrations can therefore have significant effects on the epigenome which could potentially further drive tumorigenic phenotypes, as will be discussed.

As mitochondria are key central functional hubs within the cell, reprogramming of metabolic circuits and pathways in tumor cells has far-reaching consequences, including disrupted energetic pathway or dependencies and alterations to the epigenomic landscape. Such varied and far-reaching consequences of cellular metabolism only add complexity when studying altered metabolics of tumor cells.

## 19. Epigenetics and Epigenome

Apart from DNA nucleotides (A, C, T, G), there exists an additional layer of information encoded in the nucleotide bases [442]: the epigenome. The epigenetic layer enables genomic regulation without necessitating coding sequence changes [443,444]. Epigenetic PTMs can include chemical alterations to DNA, such as the addition of methyl or acetyl groups, or alternations to the histone tails, such as (but not limited to) deposition of methyl, acetyl, or phosphoryl groups [445]. For each of these reactions, a series of protein families are employed capable of “writing”, “reading”, and “erasing” said PTM marks (Table 3). Depending on the chemical nature, PTMs possess different lifetimes. While phosphatases readily reverse phosphorylation [446,447,448], tri-methylated lysine modifications can persist for extended periods of time [448,449,450]. These modifications are utilized by the cell to regulate gene expression and chromatin architecture, which can have significant impacts on the development and function of the cell in response to internal/external stimuli or in diseased states such as cancer. Investigations have shown that many of these PTM deposition and/or excision reactions require metabolites as cofactors [451,452]. This, therefore, intimately couples cellular metabolic and epigenomic states owing to the shared generation and use of cofactors and metabolites. As such, a cell’s metabolic state can drastically shape and alter its epigenetic state and epigenetic alterations can propagate and alter a cell’s metabolic state [451,452,453,454].

## 20. DNA Packaging and Histones

Within the nucleus, DNA can take on states of euchromatin or heterochromatin [455]. Euchromatin, or an open state of chromatin, enables transcription and gene expression. Conversely, heterochromatin is tightly packed and condensed DNA around histone proteins. For this condensation to occur, a 147 bp stretch of DNA is wrapped around a core of eight histone proteins (2*(H3/H4) dimers linked to 2*(H2A/H2B) dimers), with a stretch of 20–90 bp of DNA linking adjacent histones (linker DNA) [443,456,457,458,459,460]. Histone H3/H4 dimers occupy the core of the nucleosome, while H2A/H2B dimers are more loosely associated. Histone proteins have a globular domain with a characteristic alpha-helical “histone-fold” arrangement [461,462]. The histone octamer has a positively charged surface interacting with the negatively charged DNA backbone [463]. Physically, DNA has a double-helix structure and is wound in a right-handed orientation which is then wound in a left-handed orientation around histone octamers. Owing to the size of the DNA molecule and histone proteins, the DNA molecule only winds approximately 1.65 times around a histone octamer which results in a spatial arrangement where sites 70 bp along the DNA molecule are in proximity to each other on the apical surface of the wound nucleosome [460,464,465].

Nucleosome structure varies due to differences in histone composition, which itself can show tissue-specific differences [466]. Through deposition of chemical modifications onto the N-terminus of histone tails, which are readily accessible for modifications, the function of nucleosomes can be changed, thus affecting the function of the associated DNA. Histone chemical modifications described in the literature include methylation, phosphorylation, and acetylation, with other modifications such as SUMOylation, ubiquitinylation, ribosylation, glycosylation, crotonylation, and serotonylation also being described [459,467,468,469]. More specifically, histone methylation is reported to be deposited at lysine (K) and arginine (R) residues, with deposition of multiple methyl groups possible (mono-, di-, tri-methyl group deposition). Phosphorylation can occur at serine (S) residues, with acetylation and ubiquitination also potentially occurring at lysine (K) residues. The nomenclature describing these changes follows the convention of 

**<** histone protein—H2A/H2A.z/H2B/H3/H3.1/H3.3/H4 **>**

**<** single character amino acid code—K (lysine)/R (arginine)/S (serine) **>**

**<** position of the amino acid carrying the modification **>**

**<** abbreviation of the chemical modification—me1 (mono-methylation) /

me2 (di-methylation)/me3 (tri-methylation)/p (phosphorylation)/ac (acetylation) **>**

In the following sections, we will introduce DNA methylation, followed by a discussion on histone methylation, acetylation, and phosphorylation. Although other histone modifications have been described in the literature, their exploration is beyond this review.

## 21. DNA Methylation

In animal and plant genomes, DNA cytosine bases can be modified with the chemical addition of a methyl group, resulting in 5-methyl-cytosine (5mC) [470,471]. In humans, the 5mC reaction is catalyzed by the activity of a DNA methyltransferases (DNMTs) and utilizes the metabolite cofactor S-adenosyl-methionine (SAM) as a methyl group donor [472,473]. Chemically, this reaction involves a modification of the 5th position of the cytosine ring by the thiol group of a cysteine, leading to the formation of a covalent bond between the cytosine and the DNMT protein. Once the 5mC methyl group is attached, it can be physically located in the major groove of a DNA helix where it can be accessed by DNA methylation readers, the methyl-CpG-binding domain (MBD) protein family (Table 3) [474]. De novo methylation is described to occur by the DNMT3 family of proteins, namely DNMT3A/B, with DNA methylation maintenance being performed by DNMT1 [442,475,476,477,478,479] (a detailed review of the DNMT family can be found in reference [480]). Despite these canonical roles, evidence has emerged that each protein is capable of having a compensatory function for the other DNMTs should the cell require it; i.e., DNMT1 can show de novo methylation function, while DNMT3A/B can show DNA maintenance function [481,482]. Such compensatory behavior is predicted to occur in disease states, such as cancer, where mutations can result in loss of enzymatic function of particular family members [483,484].

In mammalian genomes, 5mC is preferentially found in the context of CG dinucleotides. This CG dinucleotide sequence is palindromic across both sense and anti-sense strands, representing a symmetrical arrangement of nucleotides. During replication, when methylated DNA is copied, the synthesized (daughter) strand does not contain the methylation of the parental strand, resulting in hemi-methylated DNA [478,485,486]. Following synthesis, DNMT1 is recruited to the daughter strand to restore the methylation pattern read on the parental strand [485]. If for any reason DNA methylation maintenance is inhibited at this stage, owing to mutation, molecular inhibition, or metabolite insufficiency, the integrity of the methylation signature can be diluted or lost if this persists for subsequent cellular divisions [473,480,487].

Although DNA methylation is now a ubiquitously described epigenetic control mechanism, the significance and relevance of DNA methylation were questioned when initially observed [488,489]. Over the decades the regulatory role DNA methylation plays for a cell has become indisputable as well as its ubiquity in many organisms [490,491]. Evidence has accumulated describing the essential role DNA CpG methylation plays in regulating the epigenome, providing a stable gene-silencing mechanism in most contexts (in concert with chromatin architectural changes) without necessitating genomic sequence changes [471,481,492,493]. Methylation of CpG-rich “islands” (Figure 11) in the promoter regions of genes has been found to correlate with transcriptional repression, while unmethylated CpG islands correlate with active transcription [494,495,496]. However, there exists ambiguity as to whether CpG islands are the master methylation-sensitive regulatory loci controlling gene expression since studies have documented stronger correlations between CpG “shore” methylation and active transcription (CpG shores are described as the 2 kb regions flanking a CpG island) [493] (Figure 11). Additionally, only 30–70% of gene promoters contain CpG “island” regions (depending on the identification criteria and genome build), prompting the question, what about other genes without CpG islands? They too can show methylation-dependent transcription regulation [497]. These findings blur the lines as to which methylation-sensitive loci are truly and definitively capable of controlling gene expression [493,498,499,500,501,502,503]. Moving beyond gene promoter regions, evidence has been presented correlating unmethylated CpGs with active enhancers similar to gene promoters, however, in opposition to this trend, methylation within gene bodies correlates with active expression, rather than repression [494,495,496].

Depending on the site of modification, DNA methylation can facilitate or inhibit protein or transcription factor binding [504,505,506,507,508,509,510,511,512]. Examples of how different proteins interact with methylated or unmethylated DNA include the transcription factors CCCTC-binding factor (CTCF) and CCAAT enhancer-binding protein beta (CEBPB). CTCF can recognize methylated DNA, while CEBPB will recognize methylated or unmethylated DNA depending on its binding partners ATF4 and CEBPD, respectively [504,512,513]. Integrative studies such as these showcase the breadth of transcription factor (TF) binding in different contexts and how transcription can be facilitated or inhibited simply depending on the underlying methylation state of the genome even if the canonical binding motif is present.

In contrast to the methylation “writing” DNMT family, the ten–eleven translocation (TET) family is capable of oxidizing 5mC into its other chemical forms, leading to eventual demethylation or “erasing” of DNA methylation [514,515,516,517]. The TET family of proteins consists of TET1–3 which are ferrous ion (Fe^2+^)- and α-KG-dependent dioxygenases that use oxygen to oxidize the methyl group of 5mC [514,515,516,517,518]. In this reaction, 5mC is oxidized to 5-hydroxy-methyl-cytosine (5hmC), then 5-formyl-cytosine (5fC), and finally 5-carboxyl-cytosine (5caC) [514,515,516,517] (Figure 12). Through these TET-mediated oxidation reactions, a cell is able to actively and/or passively demethylate DNA [485,519]. It is pertinent to note that the intermediates and cofactors of these successive reactions are derived from other cellular processes, primarily metabolism [451,452,453,454]. Intermediates such as α-KG, O_2_, CO_2_, succinate, fumarate, and 2-HG are all metabolic factors [451,452,453,454]. As such, perturbations to the availability and concentration of these intermediates either through metabolic regulation, metabolic inhibition, diet, or environmental factors will affect a cell’s ability to regulate methylation adequately.

The last family of proteins associated with DNA methylation is the methylation “readers”; proteins capable of reading the deposited methylated residues and interpreting the signature for the cell. All of these proteins contain a methyl-CpG-binding domain. These proteins are described to mechanistically function by binding to methylated DNA and directly reducing accessibility for transcription factors, resulting in transcriptional repression [520,521].

## 22. Histone Methylation

Histones (namely H3 and H4) have lysine (K) and arginine (R) residues in their tails which can be methylated [444,522]. In contrast to acetylation and phosphorylation, methylation does not alter the charge of the histone protein [523]. Additionally, while DNA methylation is used as a repressive mark, histone methylation is able to both facilitate or repress transcription depending on the residue and the type of methylation applied. Histone lysine methyltransferase (KMT) enzymes utilize SAM as an intermediate donor to transfer a methyl group onto the lysine residues of histones and contain a conserved SET domain (with the exception of DOT1) [523,524,525,526]. Lysine residues show mono-, di-, or tri-methyl group deposition which is capable of facilitating or repressing transcription depending on position of deposition [463,522,527]. Arginine residues, on the other hand, show mono- or symmetrical or asymmetrical di-methylation [528].

Through numerous studies, general consensus profiles for histone lysine methylation have been observed (Table 4) such as active transcription being marked by tri-methylated H3K4 (H3K4me3), the 5′ end of transcribed genes being marked by di-methylated H3K4 (H3K4me2), active enhancers marked by mono-methylated H3K4 (H3K4me1), actively transcribed gene bodies marked by tri-methylated H3K36 (H3K36me3), and gene repression marked by di- or tri-methylated H3K9 (H3K9me2/3) and tri-methylated H3K27 (H3K27me3) [505,508,529,530,531,532,533,534,535]. While such individual observations are insightful regarding genome dynamics, integrating such observations can be more insightful by identifying potentially novel elements, as was carried out to identify low-abundance transcripts by overlapping H3K4me3 and H3K36me3 marks [536]. Other histone marks such as H3K79 and H4K20 show methylation-specific behavior depending on whether they are mono-, di-, or tri-methylated, which dictates whether they are associated with active or repressed expression [505,529,533,534]. One should keep in mind that not all genomic elements possess a single histone mark in isolation. There are many genomic loci that show bi- and tri-valency where they show deposition of activation and repressive histone marks simultaneously [444,505,527,529,533,534,537].

In contrast to histone methylation deposition, in order to remove histone methylation, cells utilize the histone demethylases (HDMs), also known as the histone lysine demethylases (KDMs), which consist of the LSD1 and Jumonji proteins [530,578,579,580] (Table 3). LSD1 proteins are dependent on the metabolite FAD, while the Jumonji family of proteins require Fe^2+^ and α-KG to perform their catalytic activity [578,579,580,581]. All demethylases have been shown to have high substrate specificity for their lysine target with some enzymes capable of only demethylating mono- and di-methyl targets, while others can demethylate all methylated lysine states [523].

In contrast to writing and erasing histone methylation, reading histone methylation is performed by a number of protein classes, including Tudor domain proteins, chromodomain proteins, and malignant brain tumor (MBT) proteins to name a few [582,583,584,585,586,587,588] (Table 3).

## 23. Histone Acetylation

Histone acetylation is regulated through histone acetyl transferases (HATs; acetylation writers), histone deacetyl transferases (HDACs; acetylation erasers) with bromodomain and extra-terminal proteins (BETs) acting as acetylation readers (Table 3). HATs include three protein families, the GNAT, p300/CBP, and MYST proteins [589,590,591]. Most described HATs can be categorized as type A HATs which are (mainly) nuclear enzymes and are responsible for acetylation of histones and non-histone proteins in the nucleus with implied functions in the epigenetic regulation of gene expression [592]. Type B HATs are instead cytoplasmic enzymes, specifically KAT1 and HAT4, and modify free histones in the cytoplasm after synthesis [592]. Histone acetylation utilizes a zinc- and ACoA-dependent process to transfer an acetyl group to histone lysine side chains. This, in turn, reduces lysine residue charge, which weakens the electrochemical connection between the positively charged histone tails and the negatively charged DNA backbone [523,589]. This causes dissociation of the DNA and histones, leaving the DNA exposed and readily accessible for transcription [523,589,593,594]. As such, histone acetylation is typically considered to facilitate gene expression and is critically dependent on ACoA stores and availability for its deposition reaction to occur [595,596,597].

In order to remove deposited histone acetylation, cells utilize the HDAC family of proteins. Four classes of HDACs have been described to date [598,599,600,601,602]. Class I HDACs are Rpd3-like proteins with members HDAC1–3 and HDAC8. Class II HDACs are Hda1-like proteins and are split into two subgroups. Class IIa shuttles between the cytoplasm and nucleus and regulates TF activity. Class IIa consists of members HDAC4, HDAC5, HDAC7, and HDAC9. Class IIb proteins, on the other hand, appear to predominantly have cytoplasmic roles and include the proteins HDAC6 and HDAC10. Class III HDACs are Sir2-like proteins and include members SIRT1–7. SIRT1, 6, and 7 localize to the nucleus where they affect gene expression. SIRT2 localizes to both the nucleus and cytosol and modulates cell cycle control. SIRT3, 4, and 5 localize to the mitochondria and respond to caloric restriction by switching cells to favor mitochondrial OXPHOS [603]. In contrast to other HDAC proteins, class III proteins utilize the metabolic cofactor NAD^+^ to catalyze their reactions, rather than ACoA and zinc [589,598,599,600]. Finally, class IV consists of only HDAC11. As a side note, in plants, a fifth group of HDACs exist, the HD2 family, which is not found in animals [604].

A number histone lysine residues have been reported to be sites of acetylation deposition, including, but not limited to, H3K4, H3K9, H3K14, H3K18, H3K23, H3K27, H3K36, H3K79, H4K5, H4K12, H4K15, H4K20, and H3K24 [605]. Of these reported marks, the most well studied is H3K27ac which, when observed in conjunction with H3 methylation, marks a spectrum of active/poised transcriptional states [506,509,540,606,607]. As noted, primed enhancers are marked by sole deposition of H3K4me1. When H3K4me1 marks are found in conjunction with H3K27ac, this marks active enhancers. In contrast, deposition of H3K27me3, H3K4me1, and H3K27ac marks poised enhancers. Additionally, co-occurrence of H3K27ac and H3K4me3 is typically associated with promoters and enhancers of actively transcribed genes [506,509]. H3K27ac deposition in intergenic regions is regarded by some to also mark superenhancers [606,608]. While such characterizations are indeed extremely insightful, the broad nature of histone peaks in sequencing studies means that their presence alone is not sufficient to accurately define which exact genomic bases perform the “enhancer” function. For this, more detailed characterizations are required to truly identify where, within the broad region identified by histone peaks, the true regulatory locus is actually located [609].

## 24. Histone Phosphorylation

Histone phosphorylation has been shown to play roles in regulating numerous cellular processes such as gene expression, DNA damage repair, apoptosis, cell cycle regulation, and cell division [468,610]. Owing to the chemical nature of this PTM, histone phosphorylation alters the overall charge of the histone protein which affects the structure of local chromatin [468,523]. Histone phosphorylation is reported to be “written” by protein kinases and is reported to occur at either serine (S), threonine (T), or tyrosine (Y) residues of histone tails [456,468,523]. Furthermore, the phosphorylation state of S, T, and Y residues has been shown to drastically alter the rate of nucleosome sliding and rearrangement [468,611,612,613,614].

All identified histone kinases transfer a phosphate group from metabolically derived ATP to the target amino acid side chain [523]. It has been shown that histone methylation sites H3K9 and H3K27 have subsequent S (phosphorylation) residues. It is hypothesized that phosphorylation at these “KS” domains may, in part, alter the affinity and/or accessibility of methylation readers and writers at the K residues [615,616]. In order to “erase” histone phosphorylation, cells utilize protein phosphatases [468]. Histone phosphatases have been shown to be essential for regulating DNA repair, mitosis, and apoptosis [468,617,618,619].

## 25. Deposition of Epigenetic Marks Is Dependent on Mitochondrial Metabolites

Cellular metabolism involves biochemical reactions to maintain the state of a cell. Metabolism impacts every cellular process and can be influenced by a cell’s epigenetic state, genetic state, environment, and dietary intake [472]. Cells use their metabolic networks to sense nutrient availability and then propagate this information to direct cellular compensatory behavior through signaling networks such as RTG responses. Downstream of this, chromatin-modifying enzymes rely on metabolites in order to carry out their PTMs. ACoA, SAM, and ATP are used for the acetylation, methylation, and phosphorylation of histones (and DNA where applicable), respectively. Therefore, concentrations of these metabolites affect gene expression by modulating epigenetic pathways, processes, and associated chromatin-modifying enzymes.

### 25.1. ACoA and Epigenetics

Acetyl coenzyme A (ACoA) is utilized by many metabolic and cellular reactions. As such, concentrations of ACoA reflect the cell’s energetic state [620]. ACoA is derived from enzymatic reactions involving glucose, fatty acids, and amino acid catabolism [620] (Figure 13). ACoA has two cellular pools, the mitochondrial and nuclear/cytoplasmic [595]. The inner mitochondrial membrane is impermeable to ACoA which is in contrast to the nuclear membrane which is freely permeable to it, thus linking the nuclear and cytoplasmic pools [595].

The majority of ACoA within the cell is produced and consumed in the mitochondria [621]. Within the mitochondria, ACoA is produced through the carboxylation of pyruvate to form ACoA, CO_2_, and NADH by pyruvate dehydrogenase [620]. When the cell experiences high/adequate levels of glucose, ACoA is produced by the oxidation of glucose with any excess glucose being shuttled to the cytoplasm to be stored as fat [597,620]. Conversely, in states of low-glucose concentrations, ACoA is produced as an end-product of β-oxidation of fatty acids that were previously stored in the cytoplasm [620,622,623,624]. Within the cytoplasm, ACoA levels are maintained by ATP-citrate lyase (ACL) and acetyl CoA synthetase (ACS) [625] (Figure 13). ACL converts citrate into ACoA and oxaloacetate [625]. This cytoplasmic pool of ACoA can also originate from reductive carboxylation of glutamine [623,624]. Functionally, cytosolic glutamine is converted into glutamate, which is then transported into the mitochondria and converted into α-KG while also generating citrate in the TCA cycle [623,624]. Citrate can then be exported back to the cytoplasm where it is converted into oxaloacetate and ACoA [623,624].

Once generated, ACoA is a key cofactor that is essential for HAT function and de novo fatty acid synthesis [597]. Cells must carefully balance the availability and usage of ACoA across these two critical cellular processes with dire phenotypic consequences should this balance be chronically perturbed [596,597,626,627]. It has been reported that hypoxic conditions, such as those found in/surrounding tumors, repress cell differentiation through reducing cellular ACoA levels which in turn leads to reduced global histone acetylation and chromatin accessibility [628]. This means that histone acetylation is highly dependent on cellular ACoA stores and availability [596,597,629,630]. When cellular ACoA concentrations are perturbed, rapid histone deacetylation occurs which can result in significant changes in chromatin structure and accessibility that can result in disease [595,597,626,627,630]. The opposing cellular mechanism to histone acetylation is the process of histone deacetylation. Regarding this, high intracellular concentrations of acetate, the precursor of ACoA, block HDAC function, which can lead to aberrant expression profiles and diseased phenotypes [631,632].

### 25.2. NNMT and Epigenetics

Nicotinamide N-methyl transferase (NNMT) is a cytosolic enzyme that catalyzes the transfer of a methyl group from SAM to nicotinamide, generating SAH and 1-methylnicotinamide (1-MNA) [633,634,635]. NNMT effectively links NAD^+^ and methionine metabolic pathways (Figure 14). As such, NNMT is capable of regulating the enzymatic reactions controlling both intracellular NAD^+^ and SAM/SAH. By being situated between these two critical cellular pathways, NNMT is able to exert effects on both metabolism (directly) and the epigenome (indirectly), which has consequent effects in diseased states such as cancer [633,636,637,638].

In cancers, NNMT has been observed to be overexpressed in breast [441,639], colon [639,640], bladder [641], neuroblastoma [642,643], lung [644], liver [639], and renal cancers [645]. The overexpression of NNMT in tumors has also been associated with the aggressiveness of certain tumors [633]. This is believed to be in part due to NNMT decreasing cellular SAM/SAH ratios which has consequent effects on reducing H3K4, H3K9, H3K27, and H4K20 methylation with downstream consequences on cell cycle- and cancer-related pathways [633,637,646,647]. As previous authors have noted, overexpression of NNMT serves as a “sink” for cellular SAM [633] thereby drastically reducing the cellular SAM pool and depriving HMTs of this essential cofactor, the deprivation of which is capable of stalling or inhibiting enzymatic function [648,649,650]. Additionally, in other cellular contexts, overexpression of NNMT has been shown to result in decreased cellular NAD^+^ concentrations while also decreasing expression of fatty acid oxidation pathway genes by inhibiting NAD^+^-dependent sirtuin function [634]. Therefore, although indirect, NNMT is capable of exerting enough influence to perturb and alter metabolite concentrations and epigenomic states of a cell.

### 25.3. NAD^+^ and Epigenetics

NAD^+^ is an essential metabolite utilized in numerous cellular processes. NAD^+^ levels have been shown to decrease under high-fat diets and increase with exercise and caloric restriction [651,652,653,654]. There has also been a link established between NAD^+^ levels and aging which has driven investigation into manipulating NAD^+^ concentrations in therapies aimed at disease prevention and lifespan extension therapies [651,652,654,655,656,657].

Within the cell, NAD^+^ can be synthesized from a number of dietary sources, including nicotinic acid (vitamin B3) and nicotinamide and tryptophan [653,654,658]. The NAD^+^ salvage pathway is also a key pathway for maintaining cellular NAD^+^ levels since it recycles nicotinamide generated as a by-product of the enzymatic activities of NAD^+^-consuming enzymes, the sirtuins (SIRTs), adenosine diphosphate (ADP)-ribose transferases (ARTs) and poly (ADP-ribose) polymerases (PARPs), and the cyclic ADP-ribose (cADPR) synthases [603,653,654,658]. Nicotinamide, a potential precursor of NAD^+^, also serves as an inhibitory factor of SIRTs, ARTs, and PARPs [653,654,658].

NAD^+^ serves both as a cofactor for enzymes that catalyze redox reactions and as a cosubstrate for the SIRTs, ARTs, PARPs, and cADPR synthases [651,652,657,659] (Figure 15). SIRTs, as discussed, are HDACs that remove acyl groups from lysine residues (e.g., H3K9ac, H3K14ac, and H4K16ac) on proteins in a NAD^+^-dependent manner [589,598,599,600]. Sirtuin’s function is therefore linked to cellular metabolism and is also sensitive to caloric restriction that alters NAD^+^ concentrations. PARP proteins, on the other hand, have established links with methylation rather than acetylation [660,661,662,663,664]. PARP proteins are enzymes that mechanistically catalyze the addition of ADP-ribose units from NAD^+^ donor molecules to target substrates [661]. Investigations have detailed how NAD^+^ regulates PARP function, as it is a required cofactor for the catalytic reaction to occur, with PARPs thereafter able to regulate DNA methylation and gene expression [653,658,660,661,662,663,664,665,666,667,668,669].

There are 17 PARP-related enzymes described in the literature [670]. PARP1 and PARP2 catalyze the polymerization of ADP-ribose units from NAD^+^, resulting in the attachment of either linear or branched poly-(ADP-ribose) (PAR) polymers to itself or other target proteins.

PARP1, in particular, has been linked to epigenetic and chromatin regulation [660,661,662,663,664,666,667,668,669]. PARP1 can modulate chromatin structure and can act as a transcriptional coregulator. There therefore exists an indirect link between NAD^+^ and cellular methylation control. It has been recently shown that hyper-NAD^+^ treatment of the leukemic cell line HL-60 resulted in impairment of DNMT1 methylation maintenance function for set gene loci, which resulted in hypomethylation and subsequent activation of the CCAAT enhancer-binding protein alpha (CEBPA) locus [665]. Although these findings have only been reported for a single genomic locus, when taken in conjunction with the additional literature evidence, these results strongly implicate a potentially undescribed, indirect epigenetic regulatory mechanism in which NAD^+^ is able to control DNA methylation through modulation of PARP proteins. Owing to the ubiquity of NAD^+^ within the cell, it therefore is plausible that cells utilize NAD^+^ for both HDAC and DNMT regulation through differing control mechanisms.

### 25.4. SAM and Epigenetics

SAM is a cofactor of DNA and histone methyltransferases and is the second most common enzymatic cofactor after ATP [648]. The methyl group of SAM serves as a donor for methyltransferase reactions as it is converted into SAH. SAH is a potent inhibitor of all methyltransferases [648,649,650]. The SAM/SAH intracellular ratio, and its control over DNA methylation and histones, is an example of how a metabolite, which can be controlled by diet and nutrition, is capable of affecting the epigenome by controlling epigenetic modifier function. SAM is produced by one-carbon metabolism through a combination of the folate cycle and the methionine cycles [648,649,650].

Enzymes that use SAM as a cofactor are DNMTs, KMTs, and PRMTs [472,473,523,524,525,526] (Figure 16). Crosstalk between metabolism and chromatin is therefore regulated by each enzyme and the physiological concentrations of its required metabolites. This is of particular importance for methyltransferases which respond to low physiological concentrations of metabolic substrates with limited and/or reduced enzymatic activity [648,649,650]. Competition for available SAM may regulate the contrasting methylation events of histone H3K4, that is associated with transcription activity, and methylation of DNA and histone H3K9, that is associated with transcriptional silencing [671,672]. Thus, changes in one-carbon metabolism, and hence alterations in the levels of SAM, can have an impact on gene expression [672,673].

### 25.5. FAD and Epigenetics

FAD is a cofactor required in many cellular oxidative reactions such as mitochondrial fatty acid β-oxidation and respiratory metabolism and is most notably required as a cofactor for lysine demethylase 1 (KDM1 or LSD1) enzymatic function [674,675,676,677,678] (Figure 17). Intracellularly, riboflavin (vitamin B2) is converted into flavin mononucleotide (FMN) followed by FAD in the cytosol [679,680]. Cytoplasmic FAD is them transported into the mitochondria where it is reduced to FADH2 in the TCA cycle by SDH as it requires FAD to catalyze the oxidation of succinate to fumarate [681,682]. In this reaction, FAD is used as a proton accepter instead of NAD^+^. The converse oxidizing reaction of FADH2 is also carried out can by reoxidized oxygen, resulting in production of FAD and peroxide to replenish cellular FAD concentrations [681].

Lysine-specific demethylate 1 (LSD1), in complex with CoREST, utilizes FAD to demethylate mono- and di-methylated histone H3K4 residues [578,678,683,684,685]. LSD1 has also been shown to be involved in the demethylation of H3K9 [685] as well as other non-histone proteins such as p53 [686].

### 25.6. α-KG, 2-HG, and Epigenetics

α-KG-dependent dioxygenases catalyze hydroxylation reactions on diverse substrates, including proteins and nucleic acids [687]. TETs and Jumonji C family (JmjC) histone demethylases are dependent on ferrous ion (Fe^2+^) and α-KG for their enzymatic activity to facilitate the removal of methyl groups from cytosine bases and histone residues, respectively [530,578,579,580]. TET proteins have been described as DNA demethylation enzymes (Table 3), although there still exists some ambiguity regarding the exact mechanism of DNA demethylation and the degree to which active versus passive demethylation is employed within a cell. Conversely, JmJC demethylases are a group within the KDMs [688]. There are several subfamilies of JmjC KDMs that have been identified [530,578,579,580].

TETs and JmjC KDMs both require ferrous ion (Fe^2+^) as a cofactor in addition to α-KG as cosubstrate to catalyze reactions in which a singular oxygen from molecular oxygen (O_2_) is attached to form a hydroxyl group in the substrate, while the second oxygen is taken up by α-KG, leading to the decarboxylation of α-KG and subsequent release of carbon dioxide (CO_2_) and succinate [578,579,580,581,687]. While KDM and TET proteins require α-KG for their required enzymatic activity, they are also sensitive to other TCA cycle intermediates, such as succinate and fumarate, which have been shown to block and inhibit their function [235,236,689]. Cells must therefore balance metabolite concentrations in order to drive/inhibit particular enzymatic activities and maintain particular cellular states and enzymatic functions. Naive embryonic stem cells (ESCs), for example, utilize both glucose and glutamine catabolism to maintain a high level of α-KG [690]. The elevated α-KG/succinate ratio in naive ESCs promotes histone and DNA demethylation and maintains pluripotency [690]. Direct manipulation of the intracellular α-KG/succinate ratio is sufficient to regulate multiple chromatin modifications, including H3K27me3 and TET-dependent DNA demethylation, which contribute to the regulation of pluripotency-associated gene expression [690].

Enzymatic reactions that utilize 2-HG have an additional inhibitory mechanism to contend with and potentially overcome: the presence of α-KG. α-KG is an oncometabolite that is formed as a consequence of somatic mutation in IDH enzymes. Although numerous somatic mutations have been identified in IDH1/2 in tumors [691,692], there appears to be a predilection for specific binding site “hotspot” mutations (Arg 132 for IDH1, Arg 172 for IDH2) in certain diseases such as leukemia and glioma. Instead of producing α-KG, these mutations instead result in the production of 2-HG, an oncometabolite that competitively inhibits αKG-dependent enzymes such as JmjC KDMs and TETs (Figure 18) [372,379,381,691,692,693,694,695]. Structurally, 2-HG and α-KG are similar with the exception of the 2-ketone group in α-KG being replaced by a hydroxyl group in 2-HG. Structural analysis showed that 2-HG occupies the same space and binding orientation as α-KG in the active site of histone demethylases, thereby preventing the binding of α-KG to the active site of the enzymes, thus it is a competitive inhibitor of 2-HG [696,697]. 2-HG also strongly inhibits histone demethylases [379]. The strongest inhibition was observed with KDM4A, which demethylates H3K9 and H3K36 [698], followed by the H3K9/H3K36 demethylase KDM4C [379] and the H3K36 demethylase KDM2A [699]. Human tumors expressing IDH1 and IDH2 mutations had increased histone methylation at H3K4, H3K9, H3K27, and H3K79 [379]. 2-HG inhibits TET activity and this inhibition could be overcome by the addition of α-KG [375]. Mutations affecting TET2 and IDH1 are mutually exclusive in AML, suggesting that their biological effect is similar and that they have overlapping roles in AML pathogenesis [377,700].

The evidence presented thus far has discussed the mitochondria and the plethora of cellular processes which they can control and direct independent of nucleus input. Evidence has also been presented discussing how perturbation to mitochondrial function, be it due to environmental factors, nutrition deprivation, molecular inhibition, or mutation (somatic or germline), can trigger numerous cellular cascades and compensatory mechanisms in order to preserve cellular survival, the most important of which is an RTG response. An RTG response triggers a cascade of changes that upregulate numerous cellular pathways and changes (such as substrate-level phosphorylation and HIF) which, if left uncorrected, drive the cell to a persistent tumor phenotype that displays aberrant growth profiles and has a resistance to apoptotic signals in conjunction with aberrant execution of other cellular processes from which it may not be able to recover should these triggering signals dissipate or subside.

An (un)intended consequence of disrupted mitochondrial function, however, is the effects it has on the cell’s epigenome. Perturbations to metabolite production, concentrations, or even the production of oncometabolites owing to somatic mutation all have inhibitory effects on the function of epigenome maintainers (Figure 19). This affects the cell’s ability to regulate its epigenome to maintain normal states and results in dysregulation of DNA methylation as well as histone methylation, acetylation, and phosphorylation profiles (Figure 19). If the epigenome is held captive in an aberrant state for an extended period of time, the cell may lose its ability to recover to a pre-perturbed state even if triggering signals are removed. How deleterious the consequent dysregulation is depends on the epigenetic mark in question and the duration of aberrance as each mark has its own duration of persistence. It therefore becomes evident that repeat incidences of mitochondrial and epigenetic dysregulatory episodes are required before cells are pushed into a chronic and persistent tumorigenic state, which can require numerous perturbation episodes over a number of years. This therefore might be why the presentation of many cancers is in older members of the population.

Taken together, the evidence clearly describes how intimately linked cellular metabolism and the epigenome truly are. The triggers that perturb and alter one will undoubtedly alter and perturb the other. Tumors are therefore critically dependent on these two cellular systems and their coupling in order to maintain tumorigenic phenotypes to support and drive essential metabolite generation, reprogrammed transcriptional circuits, reconfigured epigenetic profiles, apoptotic resistance, and essential cellular growth circuits that are typically tightly controlled and regulated by said cellular systems.

## 26. Therapies Targeting Dysregulated Metabolics and Epigenetics in Cancers

Through decades of research, evidence has been accumulated as to how essential dysregulated metabolic and epigenomic profiles contribute to continued tumor cell persistence and survival. This naturally has led to development of small-molecule inhibitors aiming to target these tumor cell dependencies for various therapies. What follows is a brief summary of compounds in development along with their intended cellular targets (Table 5).

DNA methylation regulators, as well as histone methylation and acetylation regulators, can be controlled by mitochondrially derived metabolites. By carefully orchestrating and controlling metabolism, tumor cells can drive whichever genetic program they require to drive their growth and survival, promoting hyper- or hypophenotypes depending on the required state and/or phenotype.

The data presented show numerous critical pathways and enzymes can be targeted, especially those upregulated or mutated in various cancers. Many of the compounds listed still require years of development and possibly refinement before they make their way to the clinic, but preliminary data, at least for some compounds, show clear efficacy for targeting the metabolic and epigenetic abnormalities frequently observed in tumors.

## 27. Concluding Remarks

This review aimed to summarize the body of knowledge gained investigating cellular metabolism and the epigenome. We firstly discussed the mitochondria’s essential roles and how they regulate cellular function. Next, we discussed how mitochondrial dysfunction relates to and drives key tumor hallmarks and included a detailed discussion of the altered metabolic profiles seen in cancers. These discussions outlined how numerous tumor cell characteristics are driven by dysregulation of metabolic circuits.

Next, we discussed the cellular epigenome. Owing to the breadth of PTMs possible, we focused only on methylation (DNA and histone), acetylation (histone), and phosphorylation (histone). In this review, we aimed to outline how cellular metabolism and epigenetics intersect by connecting which pathways and metabolites are linked, how they are linked, and the consequence of how metabolic dysregulation affects epigenomic regulation. We aimed to discuss how coupled and intertwined these two cellular circuits are. This coupling is fortunate for researchers since one need only identify a targetable pathway step for therapy development. As research continues and better therapies are developed, it is foreseeable that effective combinatorial and personalized treatment regimens will be developed for cancer patients which include a component targeting tumor metabolic–epigenetic dependencies and dysregulation.

## Figures and Tables

**Figure 1 biomolecules-13-00944-f001:**
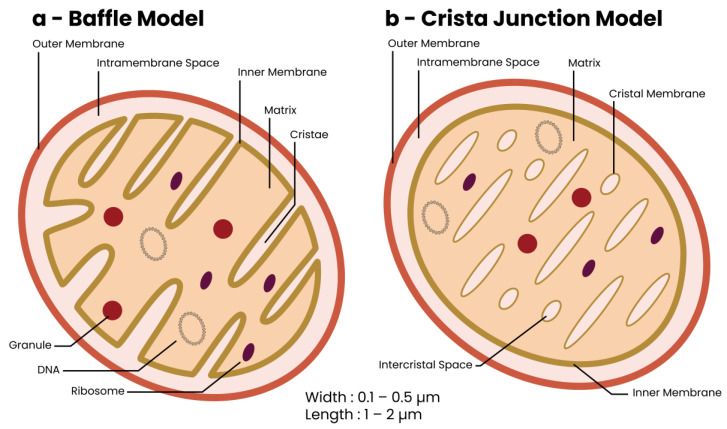
Internal structure of a mitochondrion. The mitochondrion is a double-membrane organelle with an outer membrane (OMM), an inner membrane (IMM), and numerous stacked independent membranous lamellae named cristae. Two models have been proposed describing the internal structure of a mitochondrion, (**a**) The baffle model sees cristae as random in-folds of the IMM with in-folds extending inwards towards the center of the mitochondrion. (**b**) The crista junction model sees cristae as independent membranous lamellae. These can be considered as “bubbles” within the inner mitochondria. EM micrographs have shown that the internal structure of a mitochondrion is a hybrid of both proposed models as has been shown in the literature [39,40]. Those interested are urged to view the EM micrographs in the respective publications to appreciate the internal structure of the mitochondria.

**Figure 2 biomolecules-13-00944-f002:**
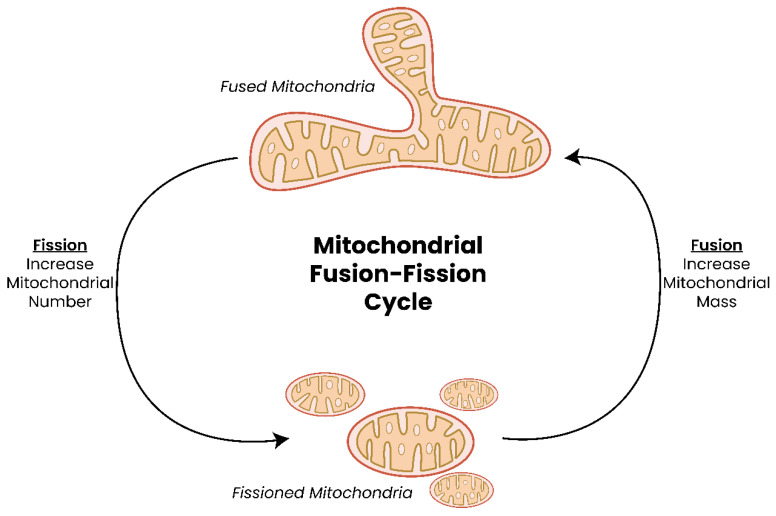
Mitochondrial fusion and fission. Mitochondria readily fuse to form large extended networks to produce greater amounts of ATP under conditions of stress or replication. The extended network is then able to break apart through the process of fission, giving rise to individual mitochondria once more. If a mitochondrion is identified to be extensively damaged or experiences sustained depolarization, it undergoes lysosomal degradation; a process known as mitophagy.

**Figure 3 biomolecules-13-00944-f003:**
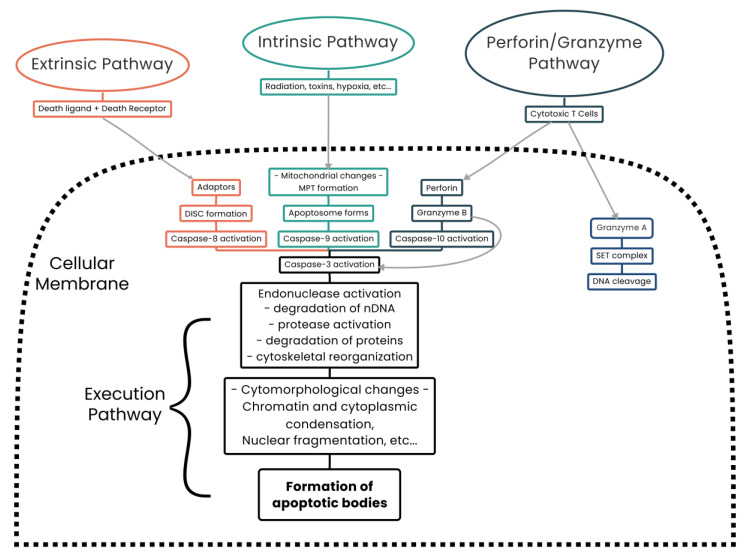
The cellular apoptotic pathways. Three apoptotic pathways are defined in the literature—intrinsic, extrinsic and perforin/granzyme-mediated apoptosis, which is unique to T-cell lymphocytes and natural killer cells. Each pathway has unique triggering conditions with all three converging on caspase-3 activation, after which the execution apoptotic pathway is common.

**Figure 4 biomolecules-13-00944-f004:**
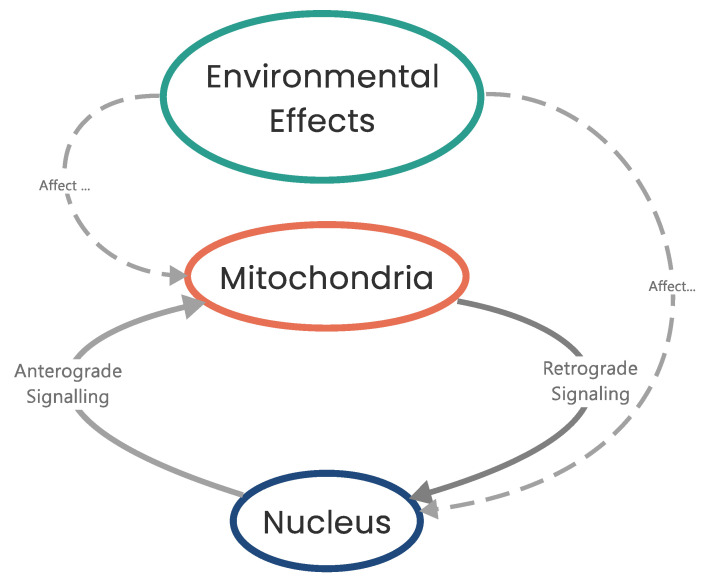
Nucleus-to-mitochondria communication pathways. Mitochondria communicate with the nucleus via retrograde (RTG) signaling pathways which are triggered on detection of mitochondrial dysfunction. The nucleus responds by anterograde signaling pathways. Environmental effects can trigger either RTG or anterograde signaling cascades.

**Figure 5 biomolecules-13-00944-f005:**
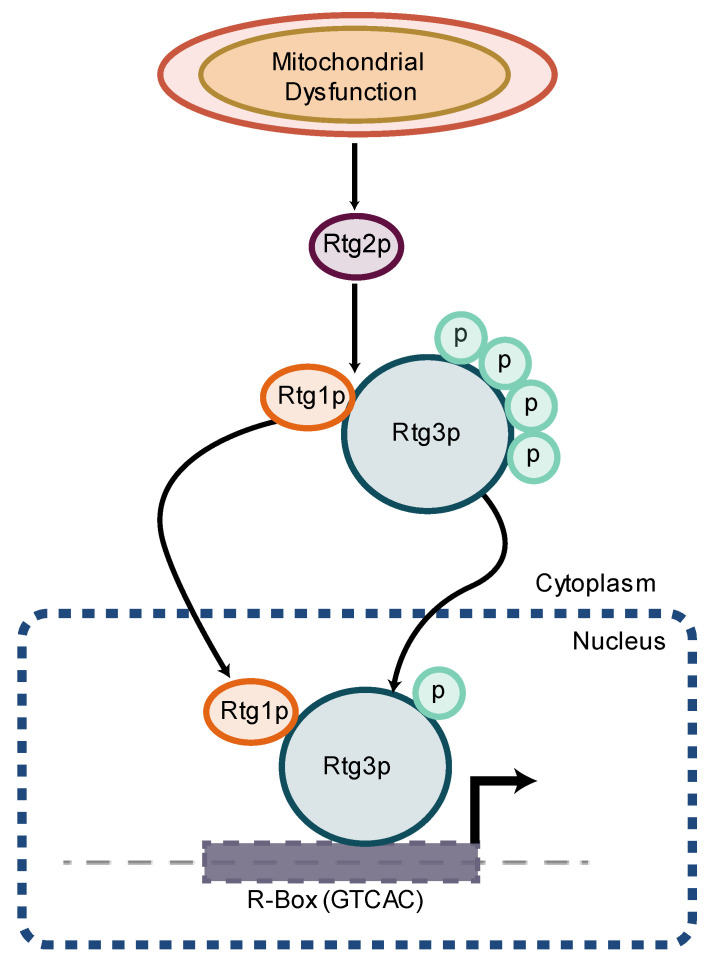
RTG response in *Saccharomyces cerevisiae.* In *Saccharomyces cerevisiae,* an RTG response results in the formation of a heterodimer between Rtg1 and Rtg3 in the cytoplasm. Rtg2 then partially dephosphorylates Rtg3, facilitating translocation of Rtg1 and Rtg3 into the nucleus. The transcription factors then bind to the “R box” sequence in the promoter region of RTG target proteins. In mammals, the Myc-Max transcription factors are homologous to Rtg1–Rtg3 and parallels have been drawn between an RTG response in *Saccharomyces cerevisiae* and an NF-κB response in mammals.

**Figure 6 biomolecules-13-00944-f006:**
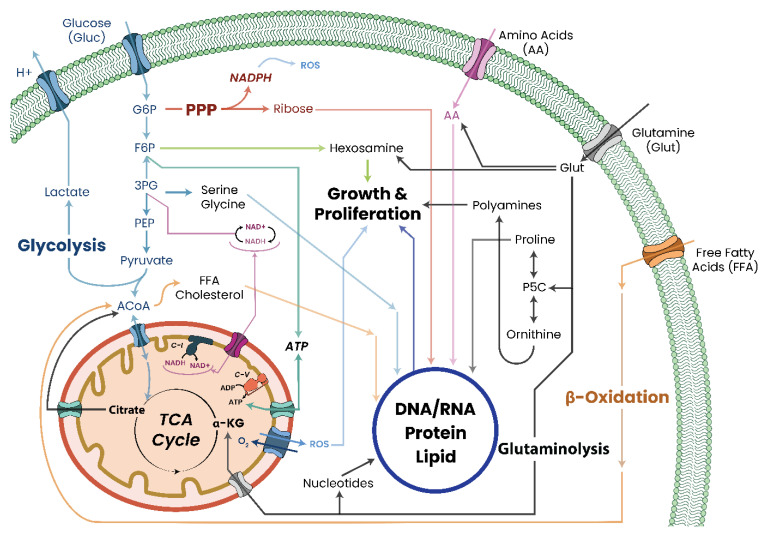
Schematic outlining the relationships between cellular metabolic pathways and the mitochondria. Although separate, each metabolic pathway within the cell feeds back into the mitochondria by producing metabolites required for ATP production. Deficiency/damage/inhibition to any of metabolic pathway can result in metabolic deficiency, potentially triggering an RTG response. Legend: **G6P**—glucose-6-phosphate; **F6P**—fructose-6-phosphate; **3PG**—glycerate-3-phosphate; **PEP**—phosphoenolpyruvate; **ACoA**—acetyl coenzyme A; **α-KG**—α-ketoglutarate; **PPP**—pentose phosphate pathway; **P5C**—1-pyrroline-5-carboxylate; **TCA**—tricarboxylic acid; **ROS**—reactive oxygen species and conditions [240]. Mitochondrial dysfunction can therefore be a major contributing factor to a varied range of pathologies with potentially serious debilitating and/or life-threatening consequences.

**Figure 7 biomolecules-13-00944-f007:**
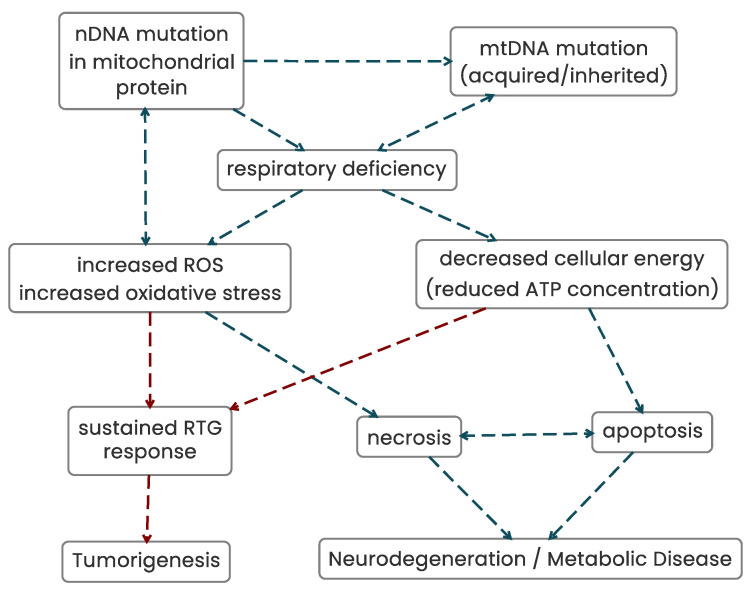
The interactions between mitochondrial mutation, dysfunction, neurodegenerative disease, and tumorigenesis. Mutations in mtDNA and nDNA result in mitochondrial dysfunction. While metabolic dysfunction typically leads to metabolic disease and/or neurodegeneration, evidence suggests that metabolic dysfunction can trigger genomic instability and trigger tumorigenesis.

**Figure 8 biomolecules-13-00944-f008:**
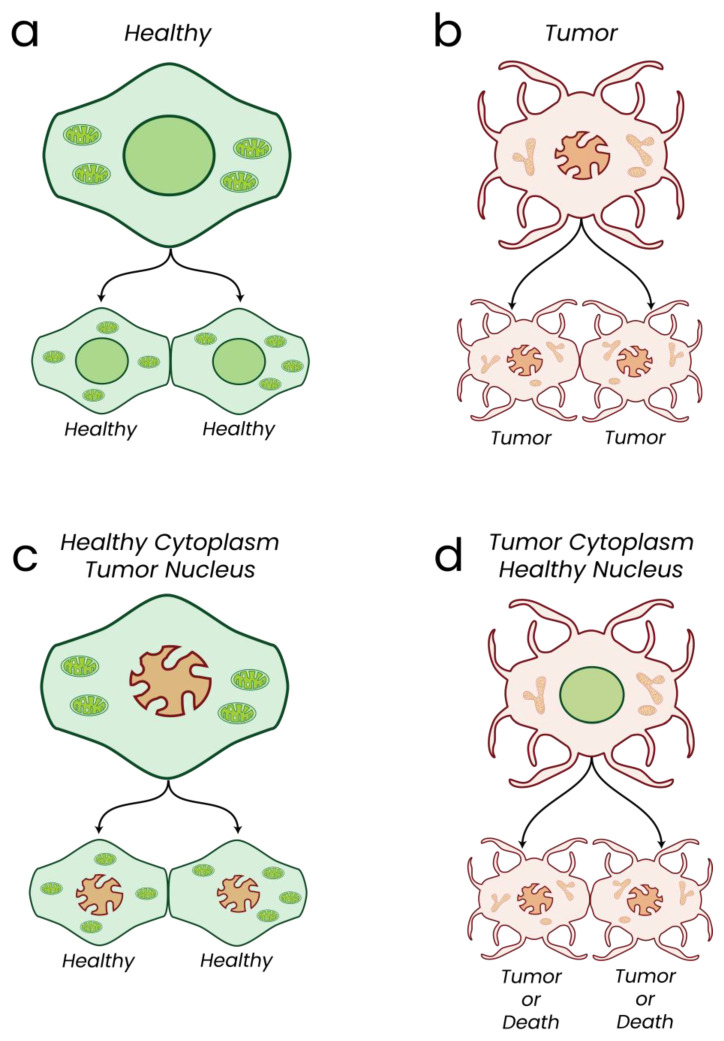
Key findings of nuclear-to-cytoplasmic transfer experiments. Nuclear-to-cytoplasmic transfer experiments highlight the significant role mitochondria play in initiation and persistence of a tumorigenic phenotype. Details of models utilized can be found in text.

**Figure 9 biomolecules-13-00944-f009:**
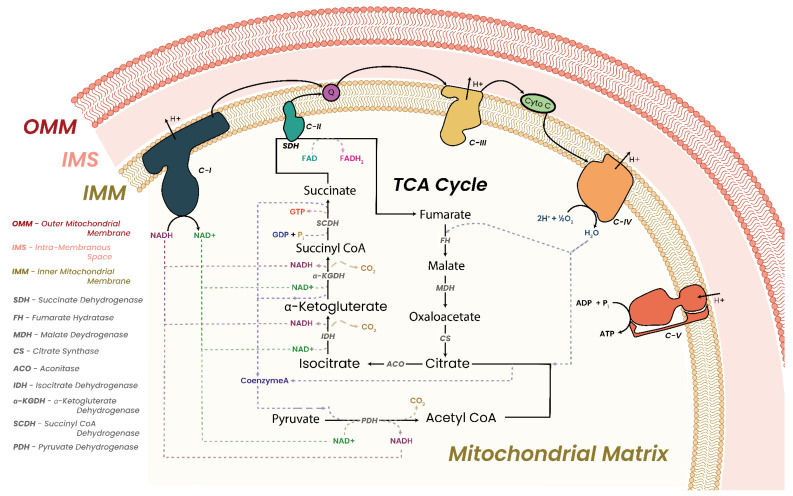
The MRC and TCA cycles. The MRC and TCA are tightly coupled. Metabolites from both pathways feed into each other to maintain function. Mutations in both the MRC and TCA have been identified in numerous cancers with established links of said mutations being tumor inducing.

**Figure 10 biomolecules-13-00944-f010:**
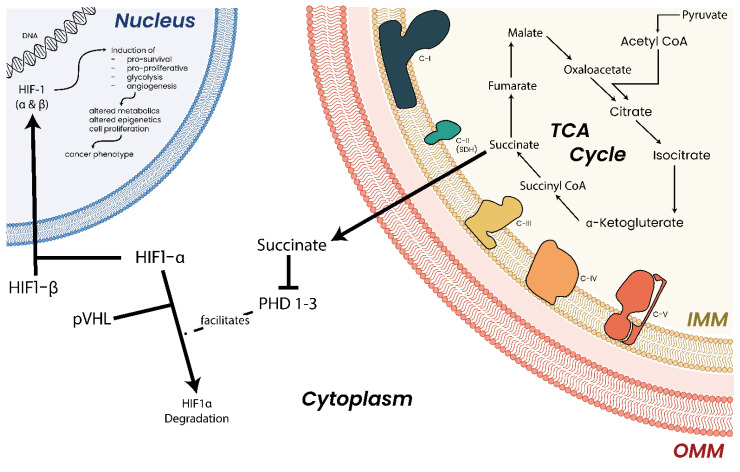
Hypothesized tumorigenic pathway consequent to succinate accumulation. C-II dysfunction leads to accumulation of the succinate. Excess succinate is transported into the cytoplasm where it inhibits prolyl hydroxylases (PHDs), thereby inhibiting HIF degradation. This leads to constitutive activation of HIF, facilitating increased expression of genes commonly associated with tumorigenesis.

**Figure 11 biomolecules-13-00944-f011:**
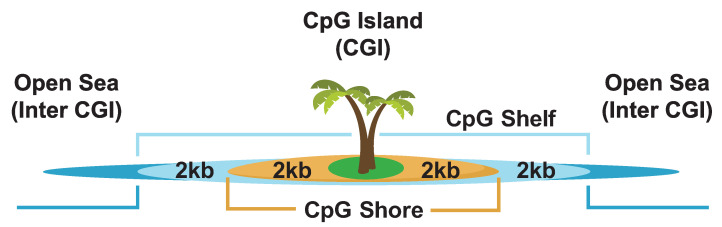
Commonly Accepted CpG Island-Centric Annotations. The commonly accepted definition of the CpG island-centric landscape situates a CpG -ich “island” in the center. Flanking the island are CpG shores, which extend for 2 kb on each flank. Thereafter, CpG shelves extend 2 kb beyond that, with any regions beyond that falling in the open sea window. The definition of what constitutes as a CpG island is dependent on the investigators’ question, what thresholds they set, and what genome build is being utilized for the definitions.

**Figure 12 biomolecules-13-00944-f012:**
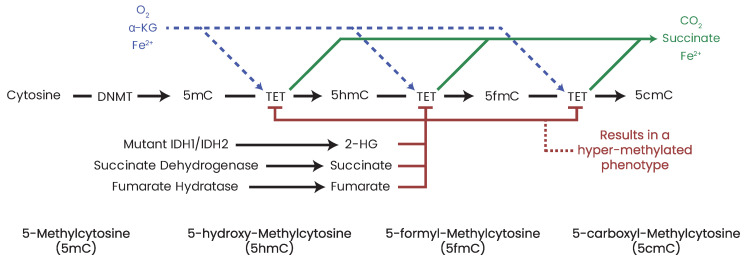
Oxidation of 5-Methylcytosine by TET Proteins. The TET enzymes sequentially oxidize 5mC to 5cmC. For these reactions, ferrous iron (Fe^2+^), molecular oxygen (O_2_), and α-KG are utilized. In cancer contexts, the production of 2-HG, hypersuccinate, and hyperfumarate contributes to inhibition of TET enzymatic function, resulting in a progression towards a hypermethylated genome.

**Figure 13 biomolecules-13-00944-f013:**
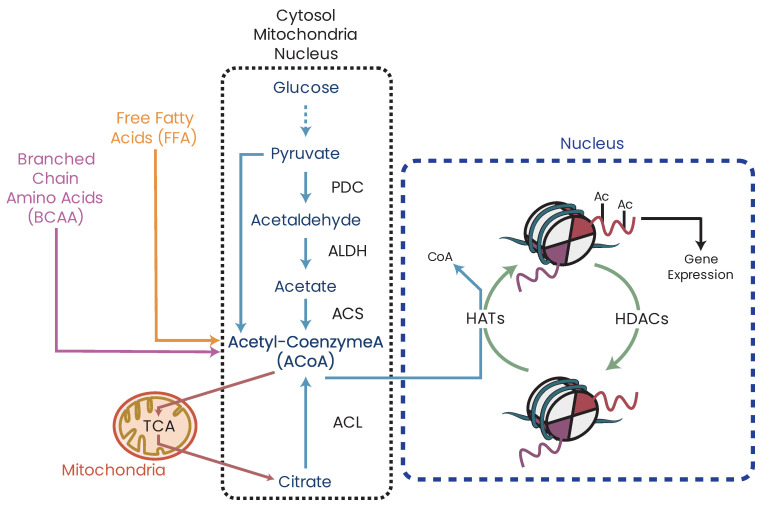
Generation of ACoA for HATs. ACoA is processed in either the mitochondria or in the nuclear/cytosolic pool. ACoA is generated primarily from glucose/pyruvate, free fatty acids, and BCAAs. Once produced, ACoA is shuttled to the nucleus where it is used by HATs as a cofactor enabling deposition of acetyl groups onto histone tails, thus enabling gene transcription.

**Figure 14 biomolecules-13-00944-f014:**
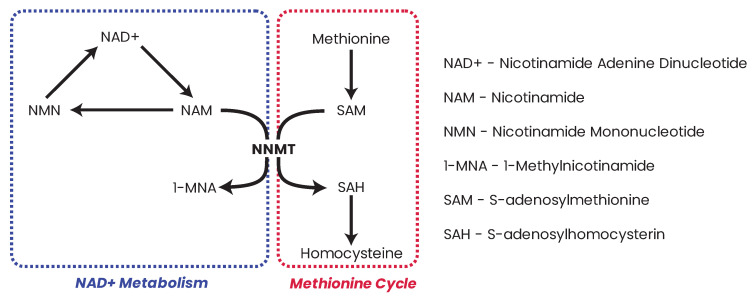
NNMT Links NAD Metabolism and the Methionine Cycle. NNMT catalyzes the transfer of a methyl group from SAM to NAM, generating SAH and 1-MNA. NAM is generated by the metabolizing of NAD^+^ by NAD^+^-consuming enzymes such as SIRTs and PARPs. Conversely, SAM is generated through the methionine cycle. NNMT therefore links NAD^+^ metabolism and the methionine cycles, with consequent effects on cellular metabolism and the epigenome.

**Figure 15 biomolecules-13-00944-f015:**
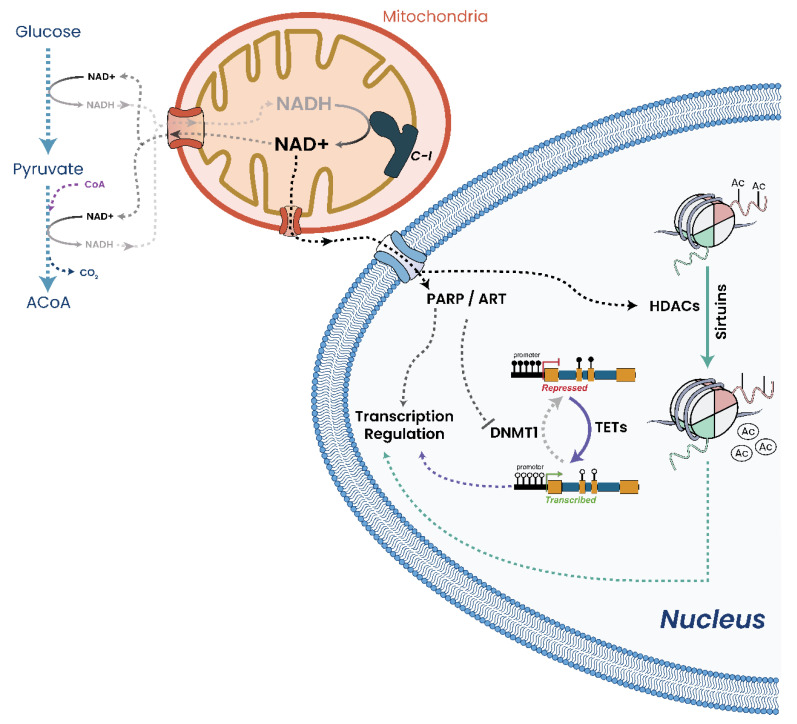
NAD’s Regulation of Histone Acetylation and Methylation. NAD^+^ is primarily generated by NADH dehydrogenase (C-I) in the mitochondria and is used as a cofactor for a number of cellular processes such as the processing of glucose into ACoA. In the nucleus, NAD^+^ is a cosubstate for histone deacetylases (SIRT family). ARTs and PARPs have established roles in regulating DNA methylation by blocking DNMT function. As such, NAD^+^ is therefore indirectly capable of regulating not only histone acetylation but DNA methylation as well. Through both of these mechanisms, NAD^+^ concentrations are capable of regulating gene expression.

**Figure 16 biomolecules-13-00944-f016:**
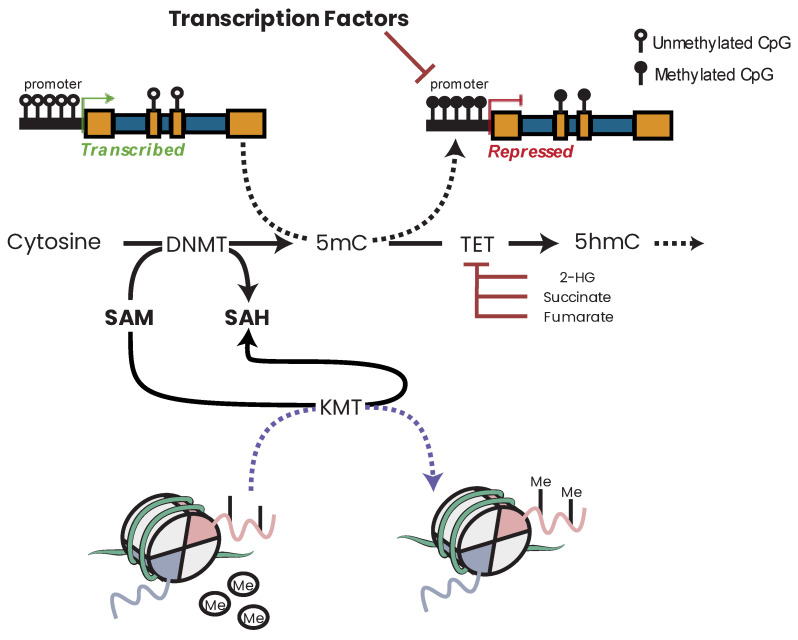
SAM as a Methyltransferase Regulator. SAM is a regulator of DNA and lysine histone methyltransferases. Cytosine is converted to 5mC by the DNMT family of proteins on DNA and utilizes the cofactor SAM as a methyl group donor. Methylated DNA results in transcriptional repression through blocking access to transcription actors. SAM is also used as a methyl group donor for the KMT family of proteins. Deposition of methylation on histone tails can have activation or repressive functions depending on the lysine residue and its position.

**Figure 17 biomolecules-13-00944-f017:**
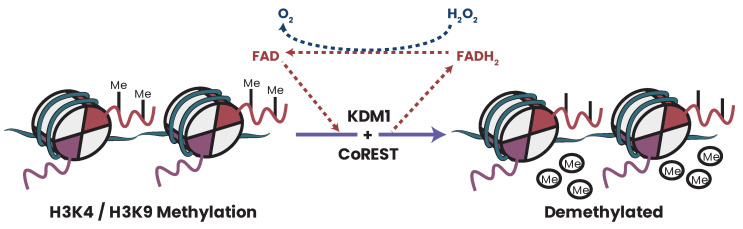
FAD and Histone Demethylation. Flavin adenine dinucleotide (FAD) is a cofactor required for lysine demethylase 1 (KDM1 or LSD1) enzymatic function. Lysine-specific demethylate 1 (LSD1), in complex with CoREST, utilizes FAD to demethylate mono- and di-methylated histone H3K4 residues, with evidence presented also supporting its role in demethylating H3K9 residues.

**Figure 18 biomolecules-13-00944-f018:**
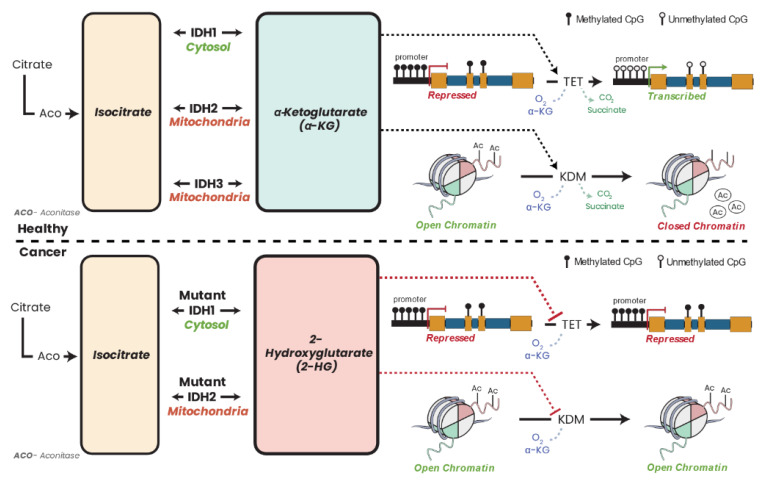
α-KG, 2-HG, and Epigenetic Control. Under healthy conditions (upper panel), the cytosol and the mitochondria IDH enzymes produce the metabolite α-KG which is utilized as a cofactor in DNA demethylation and histone deacetylation reactions. In cancer conditions (lower panel), the IDH enzymes instead produce the oncometabolite 2-HG, an α-KG analog which competitively inhibits DNA demethylases and histone deacetylases. Relationship between altered metabolism and an altered epigenome.

**Figure 19 biomolecules-13-00944-f019:**
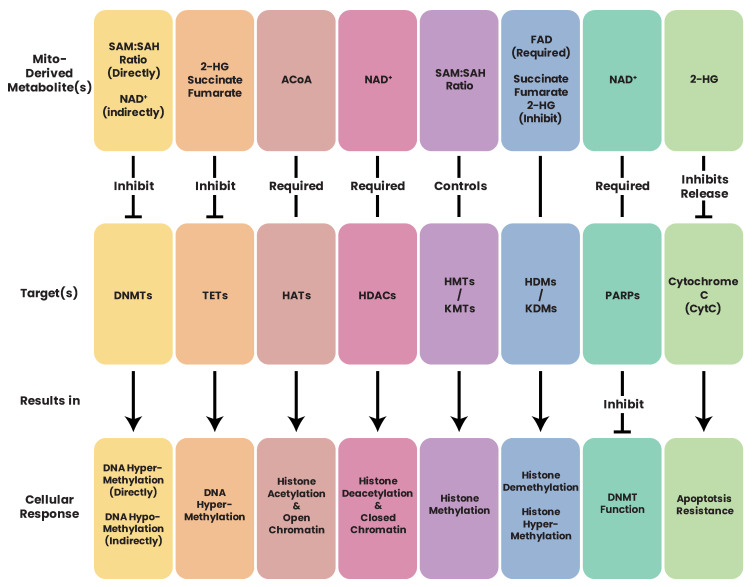
Summary of the Metabolic Regulators of Epigenetics.

**Table 1 biomolecules-13-00944-t001:** Selected somatic mtDNA mutation frequencies in cancers.

Cancer	Somatic Mutation Frequency (%)
Skin	16–75
Head and neck	49–78
Thyroid	23–100
Breast	30–93
Lung	43–79
Esophageal	5–55
Gastric	18–81
Colorectal	16–70
Pancreatic	16–92
Liver	40–68
Renal	27–79
Urinary Bladder	64–100
Prostate	19–88
Ovarian	20–80
Endometrial	9–63
Cervical	38–90
Nervous system	<35
Hematological	30–50
Connective tissue	<70

Summary of selected somatic mtDNA mutation frequencies across numerous cancers.

**Table 2 biomolecules-13-00944-t002:** Select somatic C-I variants identified in cancers.

Cancer (Reference)	C-I Gene Mutations Identified
Acute Myeloid Leukemia [31]	NDUFA12—3′ UTR—somatic NDUFA13—p.S57P—somatic NDUFAF2—intronic e2-16687—somatic NDUFS4—intronic e3-5101—somatic —5′ UTR—somatic NDUFS7—intronic e2+243—somatic NDUFV2—intronic e2-1653—somatic MT-ND1—p.Y43H—somatic —p.E204—somatic —p.E204—somatic MT-ND2—p.V43A—somatic MT-ND4—p.I8T—somatic —p.T350—somatic —p.V234—somatic MT-ND5—p.P242fs—somatic —p.L260P—somatic —p.S345P—somatic —p.S476P—somatic —p.N452S—somatic
Acute Lymphoid Leukemia [324]	MT-ND4—p.L-P variant—somatic
Chronic Lymphoid Leukemia [324]	MT-ND1—p.F-S variant—somatic
Thyroid Cancer [321]	MT-ND2—p.S-F variant—somatic —p.I-V variant—somatic —frameshift—somatic MT-ND4—p.S-F variant—somatic —p.E-K variant—somatic MT-ND5—p.L-K variant—somatic —p.S-F variant—somatic —p.I-V variant—somatic —p.D-G variant—somatic —p.A-G variant—somatic —p.S-M variant—somatic —p.I-V variant—somatic MT-ND6—p.V-A variant—somatic —p.W-R variant—somatic
Oncocytomas [303]	MT-ND1—p.G120X—somatic —p.G244X—somatic MT-ND4—p.374X—somatic MT-ND5—p.540X—somatic MT-ND6—p.87X—somatic

**Table 3 biomolecules-13-00944-t003:** Epigenetic Regulator Families.

Modification	Writers	Readers	Erasers	Genes Mutated in Cancer
DNA Methylation	DNA Methyltransferases (DNMTs)	Methyl-CpG-Binding Domain Proteins (MBDs)	Ten–Eleven Translocation Proteins (TETs)	DNMT3A DNMT3B TET1 TET2 MBD4 IDH1/2
Histone Acetylation	Histone Acetyltransferases (HATs) including GNAT, p300/CBP, MYST Protein Families	Bromodomain and Extra-Terminal Proteins (BETs)	Histone Deacetylases (HDACs)	CBP EP300 KAT6A KAT6B BRD1 BRD2 BRD3 BRD4 TRIM33
Histone Methylation	Histone Methyltransferase (HMTs)/ Histone Lysine Methyltransferase (KMTs)	PHD Finger (PHF); Chromodomain-Containing (CHD); Malignant Brain Tumor (MBT); Tudor Domain; PWWP Domain	Histone Demethylases (HDMs)/ Histone Lysine Demethylases (KDMs) including LSD1 and Jumonji-C Proteins (JHDM/JmjC)	EZH2 MLL2 KDM3A KDM4C KDM5A KDM5C KDM6A MSD1/KMT3B SETD2/KMT3A EHMT1
Histone Phosphorylation	Protein Tyrosine Kinases (PTKs)	14-3-3 Proteins	Protein Tyrosine Phosphatases (PTPs)	PTPN1 PTPN11 PTPN13 PTPRB PTPRC PTPRD

**Table 4 biomolecules-13-00944-t004:** Epigenetic Modifications and Function.

Modification	Function	Reference(s)
DNA Methylation		
CpG Islands, CpG Shores	Inversely associated with transcription	[490,492,521,538,539]
Gene Bodies	Positively associated with transcription	[495,529,540]
Intergenic	Positively associated with transcription	[541]
Enhancers	Inversely associated with transcription	[495,529,540]
Histone Modifications		
H3K4me1	Active enhancers and promoters; Poised enhancers and promoters when in conjunction with H3K4me3 + H3K27me3	[529,532,540]
H3K4me2	Active promoters when in conjunction with H3K4me3; Poised marker when alone	[535,542,543,544] [545,546,547]
H3K4me3	Active promoters and transcription	[535,548,549,550]
H3K9ac	Active promoters	[529,532,548]
H3K9me1	Active promoters	[529]
H3K9me2	Silenced promoters and transcription	[551,552]
H3K9me3	Silenced promoters and transcription	[553,554]
H3K27ac	Active enhancers and promoters	[506,555]
H3K27me1	Active transcription	[556,557,558]
H3K27me2	Active transcription	[558]
H3K27me3	Silenced promoters and transcription	[559,560]
H3K36me3	Active gene bodies	[529,532]
H3K79me2	Active transcriptional elongation	[561]
H3R2me2	Counter-correlates with H3K4me3 at promoters; Enriched in gene bodies	[562]
H3R8me2	Silenced promoters	[563]
H3R17me2	Active transcription	[564]
H3R42me2	Active transcription	[565]
H4K20me1	Active enhancers and promoters	[566]
Variant Histones		
H2A.X	DNA repair site	[567]
H2A.Z	Active promoters and DNA repair sites	[568,569]
H3.3	Active transcription	[570]
Non-coding RNA		
miRNA	Repression of gene expression	[571,572,573,574]
lncRNA	Gene expression regulation	[575,576,577]

**Table 5 biomolecules-13-00944-t005:** Select Metabolic and Epigenetic Inhibitors Being Tested or Used in Cancer Therapies.

Drug	Target	Mode of Action	Epigenetic Consequence	Treats	References
5-Aza-2′-deoxycytidine (Dacogen/Inqovi)	DNMTs	Integration into DNA to block DNMTs	Induces hypomethylation	AML and MDS	[701,702,703,704]
5-azacytidine (Onureg/Vidaza)	DNMTs	Integration into DNA to block DNMTs	Induces hypomethylation	AML, MDS, CMML	[703,704,705]
3-Bromopyruvate (BrPA)	Glyceraldehyde 3-phosphate dehydrogenase	Suppress production of ACoA resulting in depletion	Reduced histone acetylation	Prostate, breast, hepatic cancers	[706]
Bis-2-(5-phenylacetamido-1,2,4-thiadiazol-2-yl) ethyl sulfide (BPTES)	Glutaminase	Suppress production of ACoA resulting in depletion Suppress 2-HG levels	Reduced histone acetylation	Breast cancer	[707]
CB-839 (Telaglenastat)	Glutaminase	Suppress production of ACoA resulting in depletion Suppress 2-HG levels	Reduced histone acetylation	Multiple cancers	[708,709,710]
Compound 968 (Glutaminase C-IN-1)	Glutaminase	Suppress production of ACoA resulting in depletion Suppress 2-HG levels	Reduced histone acetylation Reduced H3K4me3 methylation	Breast cancer	[711,712]
Zaprinast	Glutaminase	Suppress production of ACoA resulting in depletion Suppress 2-HG levels Reduces H3K27me2/me3	IDH1 mutant cancers	AML, glioblastoma	[709,713]
Butyrate	HDACs	Reduces glucose update	Restores histone acetylation balance	T-cell lymphoma, colorectal cancer	[714,715,716,717,718]
Vorinostat (Zolinza)	HDACs	Reduces glucose update	Restores histone acetylation balance	T-cell lymphoma	[719,720]
Romidepsin	HDACs	Reduces glucose update	Restores histone acetylation balance	T-cell lymphoma	[721,722]
2-Deoxyglucose (2-DG)	Hexokinases	Depletes ACoA stores	Reduced histone acetylation	Multiple cancers	[723,724]
FT-2102 (Olutasidenib)	Mutant IDH1	Suppress 2-HG production	Enables proper TET function	AML, glioma, MDS	[725]
LY3410738	Mutant IDH1	Suppress 2-HG production	Enables proper TET function	AML and MDS	[726]
DS-1001b	Mutant IDH1	Suppress 2-HG production	Enables proper TET function	Glioma	[727]
AG-881 (Vorasidenib)	Mutant IDH1 and IDH2	Suppress 2-HG production	Enables proper TET function	AML, glioma, MDS, chondrosarcoma	[728]
AG-120 (Ivosidenib)	Mutant IDH1	Suppress 2-HG production	Enables proper TET function	AML, glioma, MDS, chondrosarcoma	[729]
BAY1436032	Mutant IDH1	Suppress 2-HG production	Enables proper TET function	AML	[730]
IDH305	Mutant IDH1	Suppress 2-HG production Reduces H3K9me3 and H3K27me3 methylation	Enables proper TET function	Glioma, AML, MDS	[731]
AGI-5198	Mutant IDH1	Suppress 2-HG production Reduces H3K9me3 and H3K27me3 methylation	Enables proper TET function	AML, glioma, MDS, chondrosarcoma	[732]
GSK321	Mutant IDH1	Suppress 2-HG production	Enables proper TET function	AML	[372,733]
GSK864 (Derivative of CSK321)	Mutant IDH1	Suppress 2-HG production	Enables proper TET function	AML	[733]
ML309	Mutant IDH1	Suppress 2-HG production	Enables proper TET function	AML, glioblastoma	[734]
2-(3-Trifluoromethylphenyl) isothiazol-3(2H)-one	Mutant IDH1	Suppress 2-HG production	Enables proper TET function	Glioma	[735]
AG-221 (Enasidenib)	Mutant IDH2	Suppress 2-HG production	Enables proper TET function	AML, glioma, T-cell lymphomas, chondrosarcoma, cholangiocarcinoma	[736,737]
AGI-6780	Mutant IDH2	Suppress 2-HG production	Enables proper TET function	AML	[738]
CPI-613 (Devimistat)	TCA Cycle Intermediates	Pyruvate dehydrogenase alpha-ketoglutarate dehydrogenase	Blocks enzymatic activity	Burkitt’s lymphoma, MDS, T-cell lymphoma	Company press releases
IACS-010759	NADH dehydrogenase (C-I)	Quinone-site inhibitor	Blocks C-I function and electron transfer	Multiple cancers	[739,740]
DZNep (3-deazaneeplanocin A)	SAH hydrolase	Increases SAH:SAM ratio Degrades EZH2	Inhibits H3K27me3 and H4K20me3	Multiple cancers	[741]
Adenosine dialdehyde	SAH hydrolase	Increases SAH:SAM ratio	Inhibits DNA and histone methylation Downregulates MMP-9 and inhibits Ras/Raf-1/ERK/AP-1	Multiple cancers	[742]
N-methylnicotinamide	NNMTT	Increases SAM levels	Impairs methylation	AML, multiple cancers	NCT02746081, NCT03127735
TVB-2640	Fatty acid synthase	Blocks fatty acid processing		Solid tumors, lung, ovarian, breast	[743,744]

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
