# Peer review of "The Interplay between Dysregulated Metabolism and Epigenetics in Cancer"

_biomolecules, 2023, doi:10.3390/biom13060944_

Round 1

Reviewer 1 Report

In this article, the author discusses the role of cellular energetics or metabolism and epigenetics are tightly coupled cellular processes and the iteration of these two aspects in the process and reprogramming of tumor cells. Furthermore, the author discusses the impact of metabolic cofactors in epigenetic regulation. The author highlighted the role of the mitochondria and mitochondrial respiratory chain (MRC) and all other related actions in the tumor process.

1) I would recommend a thorough grammar check of the entire manuscript and a weighted edition of the introduction if too much information is presented non-fluently.

2) Several scientific studies have shown that the enzyme involved in the metabolism of pyrimidines, for example the upregulation of the enzyme thymidine phosphorylase induces an anti-apoptotic action and promotes proliferation in various types of cancer and consequently it has been demonstrated that the use of a drug containing tipiracil/trifluridine improves the conditions of response to therapy, for this reason I recommend expanding the bibliography with works that deal with the topic, for example: N. Zizzo et al., “Thymidine Phosphorylase Expression and Microvascular Density Correlation Analysis in Canine Mammary Tumor: Possible Prognostic Factor in Breast Cancer.,” Front. Vet. Sci., vol. 6, p. 368, 2019, doi: 10.3389/fvets.2019.00368.

Author Response

1 - The review has been read and grammatically corrected where required.

2 - Pyrimidine metabolism and its regulation is beyond the context of this review. While it is an interesting topic to explore for sure, it is beyond the (already long) scope of this review. I do not see that adding a section on pyrimidine metabolism will significantly add to the message of this review. Furthermore, a section on pyrimidine metabolism would necessitate a section on purine metabolism. Again, I see those discussions as beyond the scope of this review and so regrettably, will not be adding said discussions into the manuscript.

Reviewer 2 Report

The manuscript “The interplay between dysregulated metabolism and epigenetics in cancer” is a review article regarding the links and implications between the dysregulated metabolism and epigenetics in cancer cell.

I really appreciate the work performed by the author since the manuscript is generally well written, detailed and easy to read.

Nonetheless, there are some concerns that should be addressed in order to consider the manuscript suitable for the publication:

1.       In some paragraphs (e.g. “The mitochondrion”) there is written something awkward: “Error! Reference source not found”

2.       I think the legend of the figure 1 should contain some explanation of the two models.

3.       “saccharomyces cerevisiae” must be always written with the first letter in uppercase.

4.       The paragraphs “NAD+ and Epigenetics” and “SAM and epigenetics” ignore a master regulator of intracellular NAD and SAM content, namely the nicotinamide N-methyltransferase (NNMT) (PMID: 36829935). This enzyme has been reported to be overexpressed in a variety of solid tumors, where it contributes to the tumorigenicity and aggressiveness. Since NNMT can affect NAD homeostasis, NAD-dependent enzymes and concentration of SAM, it has a great impact on epigenetics, as demonstrated by Ulanovskaya et al. in an elegant study (PMID: 23455543). Moreover, it seems that NNMT can impact the metabolism of the cell (PMID: 37014628; PMID: 36750850)

Notably, a number of NNMT inhibitors are already available and seems to be a promising strategy for targeted therapy in cancer and for metabolic disorders (PMID: 34572571; PMID: 34704059; PMID: 34424711; PMID: 36104373).  All these considerations improve the manuscript and therefore should be included.

5.       In figure 8 the word “cytoplasm” is cut.

6.       Figure 9 is not very clear. For instance, the plus of “NAD+” seems unreadable. “oxalo-acetate” should be written “oxaloacetate”. “Acetyl Co-A” as “Acetyl-CoA”. “α-Ketoglutarate”. Some of these mistakes are present also in figure 10.

7.       The quality of figure 14 is not adequate and should be improved (left side).

8.       Some typos are present and should be fixed.

The manuscript is well written and of interest for the readers. However, there are some points that must be fixed in order to guarantee the scientific soundness of the article.

Author Response

I'd like to thank the reviewer for their detailed and thoughtful feedback on the presented review. I have endeavored to address all points raised as follows.

1 - I thank the reviewer for noting this error and it has been corrected. It was an issue with reference manger not placing the correct reference.

2 - The legend for figure 1 has been updated to provide a brief explanation of the two models in line with the main text provided.

3 - Thank you for the reviewer for pointing out these mistakes which have been corrected in the revised manuscript.

4 - I thank the reviewer for their insightful feedback and drawing our attention to NNMT and its reported roles in metabolism and regulating the epigenome. To incorporate the reviewer's excellent suggestion, I have included a new section entitled “NNMT and Epigenetics” preceding “NAD+ and Epigenetics” to briefly cover the topic.

5 - I have adjusted the figure to ensure this isn’t the case. I cannot confirm it won’t occur again when the upload system converts the revised document however. Hopefully not!

6 - I have adjusted the figures as per the reviewer’s feedback and ensured that the remainder of the figures in the review do not suffer from the same problems as noted by the reviewer.

7 - (the old) Figure 14 has been adjusted as per the reviewer's recommendations and a mistake identified has been corrected as well.

8 - The review has been read and grammar mistakes have been corrected where identified.

Reviewer 3 Report

Is the text written with GPT?

There is sentence on p.4 which is out of context:

one protein for C-III, three proteins for C-IV and two proteins of C-V53 .

There is sentence on p.11 which is out of context:

target genes (Figure 5) 196−198

Please, read text carefully as there are many such errors!  

Author Response

1 - No, ChatGPT was not used in the writing of this manuscript. The author wrote the work that they are submitting for publication.

2 - The reviewer's comment relates to sentences that have been truncated due to the positioning of figures, but it is important to note that they are not taken out of context. The identified sentences have been appropriately formatted for clarity.

Round 2

Reviewer 2 Report

The manuscript has been improved and can be published.

Reviewer 3 Report

I accept author's answers to my questions.